# Omni-Weather: A Unified Multimodal Model for Weather Radar Understanding and Generation

**Zhiwang Zhou[1,2][†], Yuandong Pu[2,3][†], Xuming He[2,4], Yidi Liu[2,5], Yixin Chen[2,6], Junchao Gong[2,3],**
**Xiang Zhuang[2,4], Wanghan Xu[2,3], Qinglong Cao[2,3], Shixiang Tang[2], Yihao Liu[2✉],**
**Wenlong Zhang[2✉], Lei Bai[2✉]**

[1]Tongji University    [2]Shanghai AI Laboratory    [3]Shanghai Jiao Tong University
[4]Zhejiang University    [5]University of Science and Technology of China    [6]UCLA

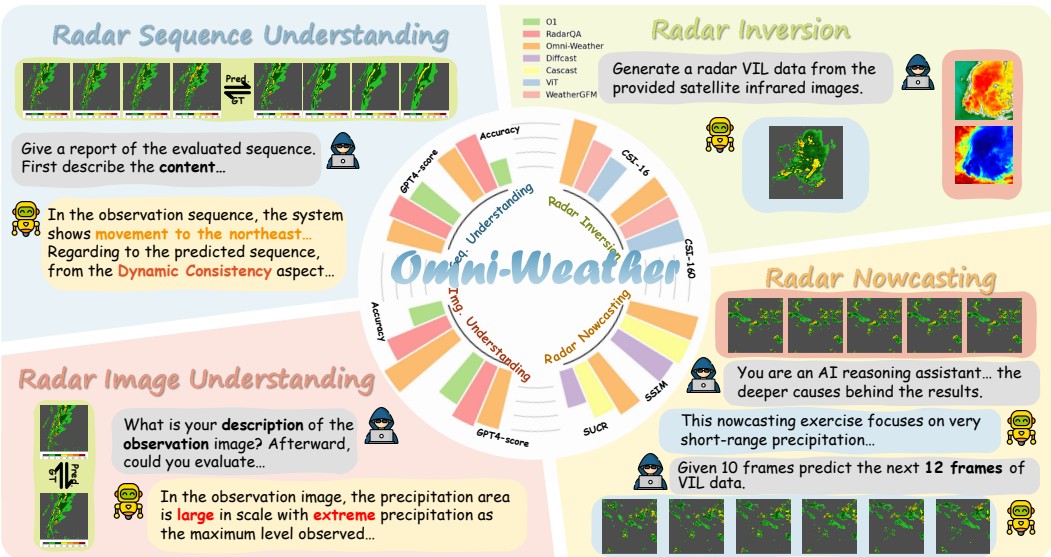

Figure 1: **Illustration of Omni-Weathers unified capabilities.**

## ABSTRACT

Weather modeling requires both accurate prediction and mechanistic interpretation, yet existing methods treat these goals in isolation, separating generation from understanding. To address this gap, we present **Omni-Weather**, the first multimodal foundation model that unifies weather generation and understanding within a single architecture. Omni-Weather integrates a radar encoder for weather generation tasks, followed by unified processing using a shared self-attention mechanism. Moreover, we construct a Chain-of-Thought dataset for causal reasoning in weather generation, enabling interpretable outputs and improved perceptual quality. Extensive experiments show Omni-Weather achieves state-of-the-art performance in both weather generation and understanding. Our findings further indicate that generative and understanding tasks in the weather domain can mutually enhance each other. Omni-Weather also demonstrates the feasibility and value of unifying weather generation and understanding. The code and dataset are publicly available at `https://github.com/Zhouzone/OmniWeather`

---

[†]Equal Contribution.

# 1 INTRODUCTION

A significant trend in AI research is the rise of foundation models that unify *generation* and *understanding* within a single architecture. Multimodal LLMs such as InternVL Chen et al. (2024c), Uni-Gen Tian et al. (2025), and Lumina-omnilv Pu et al. (2025) demonstrate that perception and synthesis can be integrated seamlessly, achieving strong generalization across visual and textual domains. These advances highlight the opportunity to extend unified generationunderstanding paradigms to weather domain, where both predictive accuracy and interpretability are essential.

Recently, weather generation and understanding tasks have made notable progress. On the generation side, nowcasting models such as PreDiff Gao et al. (2023), DiffCast Yu et al. (2024), and CasCast Gong et al. (2024) forecast convective evolution from radar sequences, supporting early warnings of hazards like flooding. Radar inversion methods He et al. (2025b) further reconstruct radar observables from satellite channels, enabling precipitation monitoring in regions without radar coverage. On the understanding side, models such as RadarQA He et al. (2025a) and WeatherQA Ma et al. (2024) generate diagnostic reports or identify severe weather impacts from atmospheric fields.

Despite these advances, unified architectures remain absent in the weather domain. As shown in Figure 2, existing approaches are divided into two disjoint paradigms: model such as ClimaX Nguyen et al. (2023) and WeatherGFM Zhao et al. (2024) excel at forecasting and downscaling but lack interpretation, while understanding models such as RadarQA He et al. (2025a) and WeatherQA Ma et al. (2024) provide diagnostic reasoning yet cannot synthesize physical fields. However, atmospheric systems are inherently multiscale, shaped by storm genesis, intensification and decay, where accurate prediction is often accompanied by the need for mechanistic interpretation. Moreover, extreme events such as rapid intensification of cyclones demand models that can not only predict hazardous outcomes but also explain the underlying drivers for actionable decision-

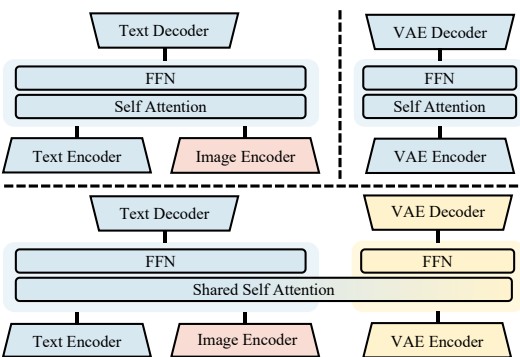

Figure 2: **Comparison** between separated architectures for weather understanding / generation (top) and unified framework with shared self-attention (bottom).

making. Current studies isolate these links*generative nowcasting models do not understand radar observations, yet MLLMs do not predict radar variables*. Bridging this gap with a foundation model that unifies generation and understanding is therefore an urgent requirement for weather domain.

To this end, we propose **Omni-Weather**, a unified multimodal foundation model for both weather generation and understanding. By consolidating these tasks within a shared backbone (Figure 2, bottom), we further propose a Chain-of-Thought dataset tailored for causal reasoning in generation tasks, which enables Omni-Weather to be finetuned with explicit reasoning supervision and to perform thinking inference. Through this integration, Omni-Weather bridges predictive accuracy with interpretability, marking a step toward reasoning unified foundation models for weather.

The main contributions of this work can be summarized as follows:

- We introduce the first multimodal foundation model in weather that jointly addresses both generation tasks (e.g., nowcasting, inversion) and understanding tasks (e.g., diagnostic reasoning, question answering) within a single model.

- We demonstrate that training both generation and understanding tasks together provides complementary supervision signals, enabling Omni-Weather to learn more transferable representations of storm evolution and improving performance on both sides.

- We propose a Chain-of-Thought (CoT) dataset and explore its integration into weather generation, enhancing perceptual quality and interpretability as a first step toward explainable generative modeling in weather domain.

## 2 RELATED WORK

**Weather generation models.** Weather generation models aim to synthesize physically consistent weather fields from historical or multi-modal observations Han et al. (2024); Chen et al. (2023); Gao et al. (2023); Yu et al. (2024); Gao et al. (2022); Lam et al. (2023). Examples include DiffSR He et al. (2025b), which reconstructs composite radar reflectivity from satellite infrared and lightning inputs, and CasCast Gong et al. (2024), which predicts precipitation evolution from past VIL sequences. More recently, foundation-scale approaches such as ClimaX Nguyen et al. (2023), FengWu Chen et al. (2023); Han et al. (2024), extend transformer architectures to climate and weather forecasting, while WeatherGFM Zhao et al. (2024) introduces in-context learning for generalist nowcasting and inversion. Despite their effectiveness in generation, these models do not address understanding or reasoning, leaving interpretability and evaluation largely unexplored.

**Weather understanding models** aim to interpret weather signals and provide human-readable insights, often through natural language or diagnostic reasoning. Early studies adapt pretrained language models such as ClimateBERT Webersinke et al. (2021); Schimanski et al. (2023) and ClimateNLP Krishnan & Anoop (2023) to analyze textual weather reports, focusing on tasks such as climate risk assessment, report classification, and domain adaptation. More recent work emphasizes multimodal inputs, combining imagery with text. For example, WeatherQA Ma et al. (2024) takes 20 images of atmospheric parameters to predict regions impacted by severe convection, while RadarQA He et al. (2025a) leverages both radar observations and numerical forecasts to generate expert-like quality assessment reports. These approaches demonstrate the feasibility of applying large language models to meteorology, but they remain limited to understanding tasks alone. In particular, existing models specialize in either textual analysis or visual reasoning without integrating predictive generation, leaving the connection between physical simulation and diagnostic interpretation underexplored.

**Unified multimodal models** integrate visual understanding and generation within a single architecture, leveraging advances in LLMs and diffusion models Chen et al. (2025a; 2024b;a; 2025b); Pu et al. (2025); Zhao et al. (2024); Zhuo et al. (2025); Ning et al. (2025). In addition to architectural unification, generalized domain prompt learning improves accessibility and transferability of scientific VLMs Cao et al. (2025). Cao et al. (2024) propose latent knowledge-guided video diffusion to generate scientific phenomena from a single initial frame. Transfusion Zhou et al. (2024) unifies text prediction and image diffusion within a single transformer trained end-to-end on both modalities. LMFusion Shi et al. (2024) adapts pretrained text-only LLMs by freezing language modules and introducing parallel image-specific branches for efficient multimodal generation. MetaMorph Tong et al. (2024) employs Visual-Predictive Instruction Tuning (VPiT) to enable LLMs to jointly predict text and continuous visual tokens from multimodal instructions. MetaQuery Pan et al. (2025) connects frozen MLLMs with diffusion decoders using learnable queries, enabling generation without compromising understanding capabilities. BLIP3-o Chen et al. (2025a) sequentially combines autoregressive modeling and diffusion to generate CLIP-aligned visual features, achieving state-of-the-art performance across modalities. BAGEL Deng et al. (2025) scales unified modeling through pretraining on interleaved text-image-video data, demonstrating emergent multimodal reasoning and manipulation abilities.

## 3 METHOD

In this section, we first introduce a unified representation of weather generation and understanding tasks, where radar nowcasting, radar inversion, and radar image / sequence understanding are formulated under a consistent sequence-to-sequence paradigm. We then present Omni-Weather, a multimodal foundation model that integrates these tasks within a shared backbone, with a detailed exposition of its architecture, modality-specific encoders, and multi-task training objectives. Finally, we describe the integration of chain-of-thought reasoning, including the construction of causal annotations and their incorporation in both training and inference, which not only enhances interpretability but also improves the perceptual quality of weather forecasts.

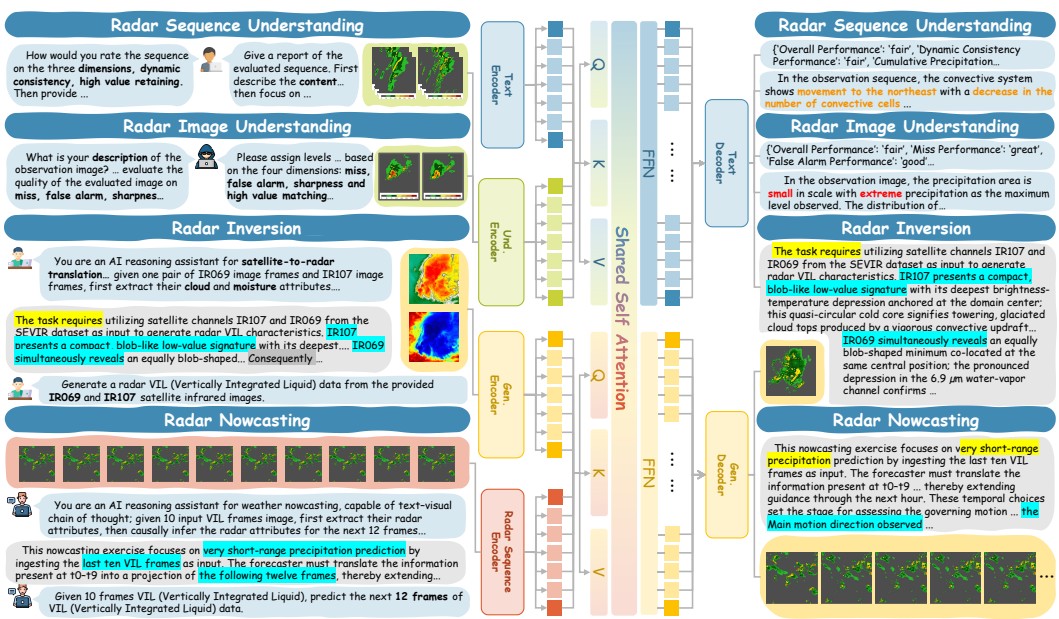

Figure 3: **Framework and Task paradigm** of Omni-Weather.

## 3.1 UNIFIED REPRESENTATION OF WEATHER GENERATION AND UNDERSTANDING TASKS

Weather modeling encompasses a wide range of objectives, from predicting future radar fields Gao et al. (2022) to generating textual assessments of forecast quality He et al. (2025a). To systematically organize this diversity, we categorize the tasks into two groups: *weather generation*, which focuses on producing future or cross-modal meteorological fields; *weather understanding*, which requires generating natural-language descriptions and evaluations. To support both categories, we leverage the SEVIR dataset Veillette et al. (2020), which provides time-aligned radar and satellite sequences of severe weather events. Below, we detail our task paradigm corresponding to each category.

**Weather Generation.** Radar nowcasting aims to predict the short-term evolution of precipitation fields. Specifically, given ten VIL frames, the model generates the subsequent twelve frames, thereby forecasting the spatio-temporal dynamics of convective systems. Radar inversion focuses on translating satellite observations into radar-derived quantities. In this task, two infrared channels (IR069 and IR107) are provided as input, and the objective is to reconstruct the corresponding VIL field, which requires a cross-modal mapping between satellite imagery and radar measurements.

**Weather Understanding.** Radar understanding tasks require the model to generate natural-language descriptions or structured assessments from radar observations and model forecasts. The input can be either a single VIL frame or a temporal sequence of frames, while the output is expected to cover key aspects, including storm morphology, intensity, temporal evolution, and forecast quality (e.g., misses or false alarms). This formulation follows RadarQA He et al. (2025a), where textual reports for frame and sequence are designed to support expert interpretation beyond traditional weather forecast metrics, which not only aligns more closely with real-world meteorological analysis, but also unifies image-level and sequence-level evaluation.

Formally, each task can be unified under a mapping $\mathcal{T} : X \rightarrow Y$, where $X$ denotes the input data and $Y$ the target output. For example, in radar understanding, $X$ corresponds to a predicted and observed VIL sequence pair, while $Y$ is a textual assessment of its quality and dynamics. This unified formulation provides a consistent modeling view across generation and understanding tasks, enabling a single architecture to learn from heterogeneous meteorological data.

## 3.2 OMNI-WEATHER: FOUNDATION MODEL FOR WEATHER GENERATION AND UNDERSTANDING

**Unified multimodal Model.** Inspired by recent advances in unified multimodal foundation models such as Bagel-7B-MoT Deng et al. (2025), we design Omni-Weather to handle both *generation* and *understanding* tasks within a single architecture. Instead of training separate models for each objective, all tasks are expressed in a consistent sequence-to-sequence format. Given a task-specific prompt $p_t$, a radar sequence input $x_t$ and target output $y_t$ (e.g., future radar sequence or radar assessment report), the model learns the mapping

$$y_t = F_\theta(p_t, x_t), \tag{3.1}$$

where $F_\theta$ denotes the shared transformer backbone. This formulation allows a single model to flexibly switch across tasks by conditioning on $p_t$ while maintaining unified training.

**Architecture.** As shown in Figure 3, Omni-Weather unifies generation and understanding within a single backbone by embedding all task prompts through the *text encoder*, thereby ensuring a shared textual space for conditioning across diverse tasks. In contrast, generation and understanding tasks have varying feature processing approches for vision modal. Specifically, For *Radar Image / Sequence Understanding*, visual inputs (e.g., a single VIL frame or a twelve-frame VIL sequence) are encoded by the *understanding encoder* and concatenated with the corresponding text prompt; the fused tokens are subsequently processed by shared self-attention layers, and the *text decoder* produces natural-language descriptions. For *Radar Inversion*, satellite channels are embedded by the *generation encoder*, fused in the shared self attention layers, and decoded by the *VAE decoder* to reconstruct the VIL field. For *Radar Nowcasting* tasks, ten input VIL frames are encoded by *radar sequence encoder*. Specifically, we instantiate this temporal encoder with EarthFormer Gao et al. (2022) to provide motion-aware tokens that stabilize long-horizon dynamics and improve temporal coherence. Since directly forcing the backbone to learn multi-frame evolution with Gen Encoder proved less stable, conditioning the shared attention layers on EarthFormers temporally aggregated tokens preserves the unified pipeline while injecting reliable temporal structure. Conditioned on the fused representation, the *VAE decoder* outputs the forecast sequence of the next twelve VIL frames.

**Training Objectives** We initialize Omni-Weather from the pretrained Bagel-7B-MoT, which provides a strong multimodal backbone trained on large-scale general data. Building on this foundation, we conduct domain-specific supervised finetuning jointly across all weather tasks.

Formally, let $\tau_t(\cdot)$ be the modality-specific encoder for task $t$. The model input sequence is defined as

$$X_t = \big[\, \tau_\text{text}(p_t)\,;\ \tau_t(x_t)\,;\ \kappa_t \,\big], \tag{3.2}$$

where $\kappa_t$ are optional conditioning tokens (e.g., temporal embeddings produced by the nowcasting encoder). The shared backbone produces contextualized tokens

$$\hat{y}_t = \begin{cases} G_\phi(f_\theta(X_t)), & t \in \mathcal{T}_\text{gen}, \\ L_\psi(f_\theta(X_t)), & t \in \mathcal{T}_\text{under}. \end{cases} \tag{3.3}$$

Here, $f_\theta(\cdot)$ is the shared encoder/backbone that produces task representations; $G_\phi$ is the VAE decoder for generation tasks $\mathcal{T}_\text{gen}$; $L_\psi$ is the text decoder for understanding tasks $\mathcal{T}_\text{under}$.

$$\mathcal{L} = \sum_{t \in \mathcal{T}_\text{gen}} \lambda_t \frac{1}{|\Omega_t|} \big\| \hat{y}_t - y_t \big\|_2^2 \ + \sum_{t \in \mathcal{T}_\text{under}} \lambda_t \left( -\sum_{k=1}^{n_t} \log p_\psi(y_{t,k} \mid y_{t,<k}, f_\theta(X_t)) \right). \tag{3.4}$$

where $\Omega_t$ indexes target pixels / frames, $n_t$ is the target text length, and $\lambda_t$ balances tasks. This SFT procedure unifies objectives under a shared representation while preserving task-specific decoding.

## 3.3 CHAIN-OF-THOUGHT CONSTRUCTION FOR WEATHER GENERATION REASONING

While unified training enables multi-task learning across generation and understanding, the resulting models still behave as black boxes, lacking explicit reasoning. To enhance interpretability and

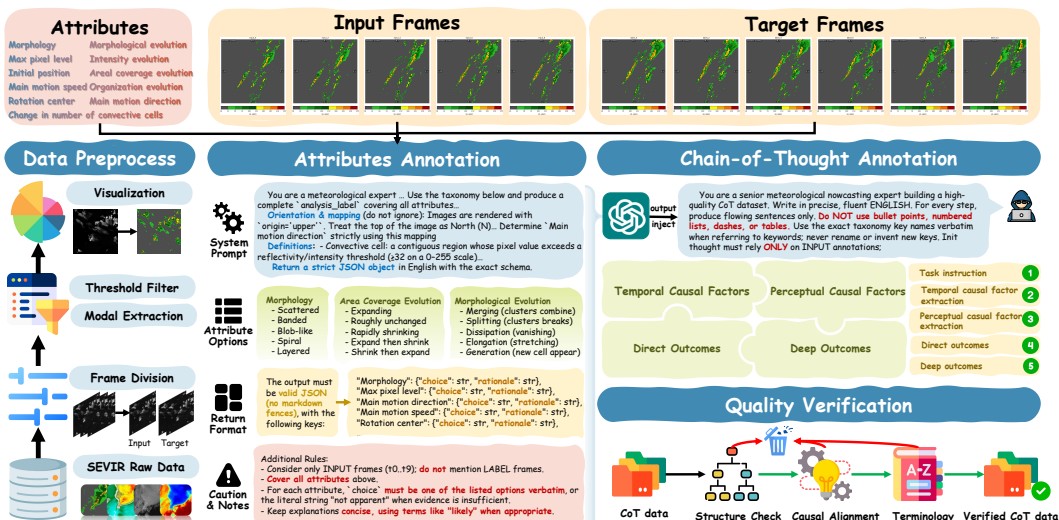

Figure 4: **Construction of our CoT data**. First, we preprocess the raw SEVIR data to obtain high-quality input / output frame pairs. Second, we carefully design prompts and leverage GPT-4o for attributes annotation. Third, the annotated attributes are incorporated into CoT prompts to generate CoT annotations, followed by a quality verification step to produce the final CoT dataset.

causal inference, we introduce *Chain-of-Thought (CoT)* supervision as an auxiliary instruction layer for generation tasks. The CoT explicitly captures causal and perceptual factors underlying meteorological evolution, thereby guiding the model toward more structured reasoning about storm dynamics.

**Chain-of-Thought for Causal Reasoning in Weather.** Our CoT formulation is tailored to the weather domain, where reasoning is framed as causal inference over storm dynamics. To operationalize this, we design a taxonomy of causal elements with expert-defined keywords, refined via GPT-based annotation. The taxonomy is adapted from RadarQA but restructured according to annotation difficulty: *causal factors* (e.g., morphology, intensity and motion) are relatively direct to extract, whereas *outcome indicators* (e.g., storm evolution patterns) require higher-level inference. For nowcasting, causal factors are first derived from the input VIL sequence, then combined with projected causal factors of the forecast frames to infer the more difficult outcome indicators describing future storm behavior. For satellite-to-radar inversion, reasoning involves only causal factors, enabling a direct projection from satellite channels to a single VIL frame. Based on this structure, we construct CoT traces in a three-stage pipeline (Figure 4): attribute annotation with GPT-4o, task-specific reasoning generation with GPT-o3, and automated verification for structural consistency, causal alignment, and terminology normalization. Detailed taxonomy design, reasoning procedures, and prompt template are provided in Appendix A.4.

**Integration for unified framework.** We incorporate CoT reasoning into Omni-Weather from two complementary perspectives. First, during the training phase, CoT serves as auxiliary supervision, requiring the model to generate both intermediate reasoning text and the final prediction, which guides the backbone toward causal interpretability. Second, during inference, CoT is utilized as a reasoning prompt, concatenated with task-specific instructions and inputs to steer the model toward more structured and explainable outputs. This integration allows CoT to not only enhance interpretability but also improve perceptual fidelity and qualitative consistency of generation tasks.

## 4 EXPERIMENTS

### 4.1 IMPLEMENTATION AND EVALUATION

We train our model end-to-end on a node with $8\times$ H200 GPUs for 20k steps, using packed sequences and the AdamW optimizer Loshchilov & Hutter (2017) with a base learning rate of $2\times10^{-4}$, weight

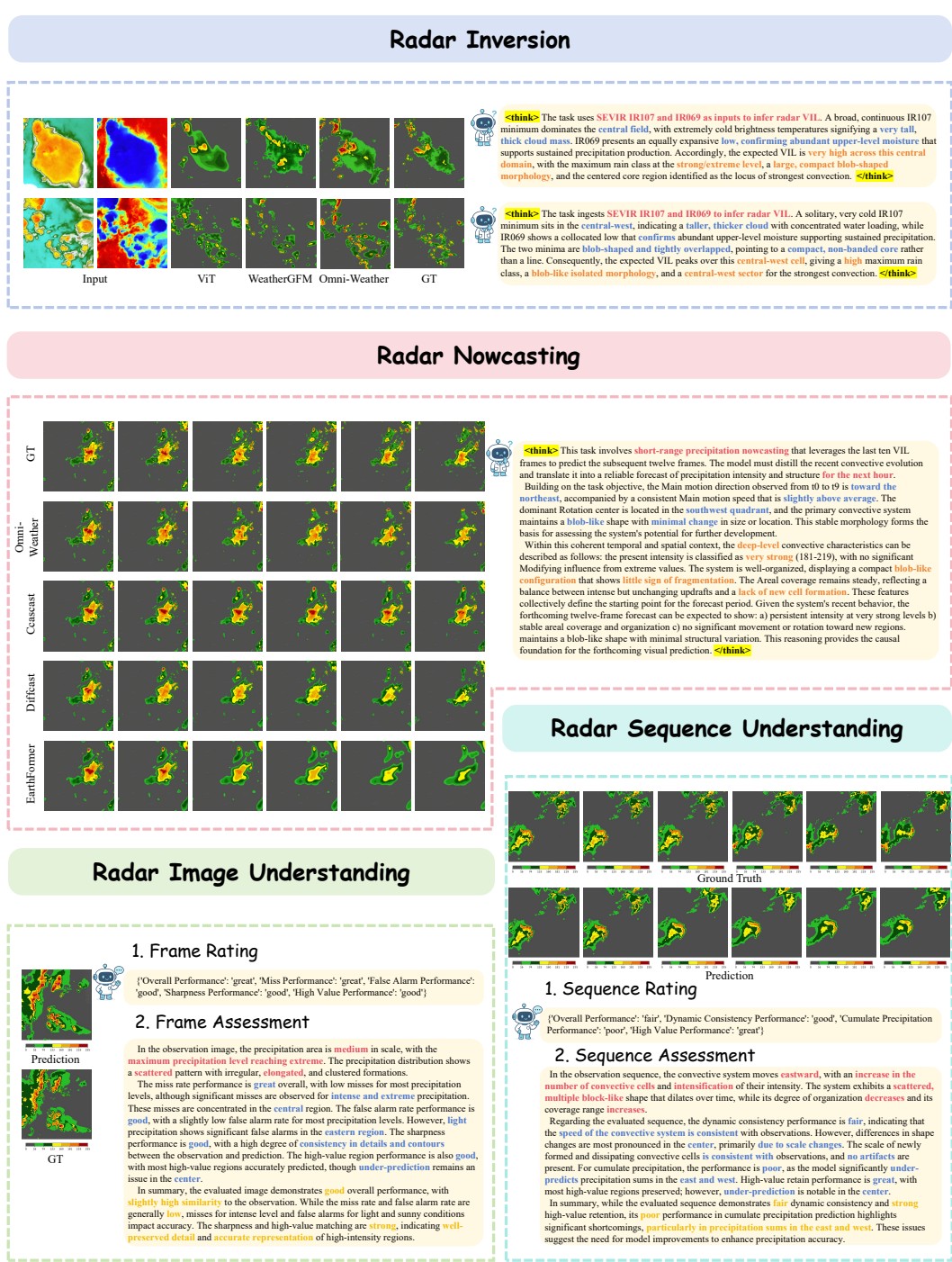

Figure 5: **A set of qualitative results.** We show two radar inversion examples with think traces, a nowcasting case where Omni-Weather (with think trace) is compared against CasCast, DiffCast, and EarthFormer, and one example each of radar image and sequence understanding with attribute scores and textual evaluations. Omni-Weather surpasses all baselines.

decay of 0.05, and cosine decay scheduling with a 2k-step warm-up. All images are capped at a resolution of $256 \times 256$, resulting in approximately 256 visual tokens per image.

For generation tasks, we report pixel-level metrics (e.g., CSI and CRPS) to evaluate radar accuracy and perceptual metrics (e.g., LPIPS and RadarQA score) to capture structural and semantic consistency. For understanding tasks, evaluation follows RadarQA protocols, considering both pre-

Table 1: **Quantitative results on Weather Generation and Weather Understanding tasks.** The best results are highlighted in bold, and the second-best results are underscored. Abbreviations: CSI-M - CSI-Mean, R.S - Radar Score, CSI-P4 - CSI-Pool4, CSI-P16 - CSI-Pool16, C-16 - CSI@16, C-74 - CSI@74, C-160 - CSI@160, C-181 - CSI@181, C-219 - CSI@219, Dyn. - Dynamic Consistency, Cum. - Cumulate Precipitation, H. Val. - High. Value, R._L - Rouge_L, B.S - BertScore, Sharp. - Sharpness. Metrics marked with ↓ denote lower-is-better objectives, whereas metrics without such notation should be interpreted as higher-is-better.

| | | | | | | | | | | | | | | | |
|---|---|---|---|---|---|---|---|---|---|---|---|---|---|---|---|
| **Weather Generation** | | | | | | | | | | | | | | | |
| Method | Radar Nowcasting | | | | | | | Method | Radar Inversion | | | | | | |
| | CSI-M | R.S | CSI-P4 | CSI-P16 | CRPS ↓ | SSIM | LPIPS ↓ | | R.S | RMSE ↓ | C-16 | C-74 | C-160 | C-181 | C-219 |
| Earthformer | **0.389** | 1.92 | 0.401 | 0.387 | 0.037 | 0.729 | 0.322 | UNet | 1.75 | 0.821 | 0.222 | 0.370 | 0.180 | 0.153 | 0.079 |
| Diffcast | 0.375 | 2.43 | 0.407 | 0.511 | 0.033 | 0.739 | 0.235 | ViT | 2.01 | 0.445 | 0.602 | 0.436 | 0.180 | 0.131 | 0.042 |
| Cascast | 0.384 | 2.72 | 0.414 | 0.518 | 0.031 | 0.746 | 0.207 | WeatherGFM | 2.28 | **0.436** | 0.619 | 0.447 | 0.208 | 0.157 | 0.053 |
| Omni-Weather | 0.384 | 2.69 | **0.427** | 0.539 | **0.026** | 0.746 | 0.179 | - | 2.42 | 0.514 | **0.622** | 0.469 | 0.263 | 0.221 | 0.118 |
| Omni-Weather-thinking | 0.353 | **2.86** | 0.421 | **0.542** | 0.028 | **0.751** | **0.166** | - | **2.51** | 0.507 | 0.621 | **0.473** | **0.277** | **0.230** | **0.129** |
| **Weather Understanding** | | | | | | | | | | | | | | | |
| Method | Radar Sequence Understanding | | | | | | | Radar Image Understanding | | | | | | | |
| | Overall | Dyn. | Cum. | H. Val. | R._L | B.S | GPT4 | Overall | Miss | FAR | H. Val. | Sharp. | R._L | B.S | GPT4 |
| Claude-sonnet-4 | 20.79 | 20.79 | 20.79 | 21.78 | 0.287 | 0.745 | 5.73 | 32.79 | 32.56 | 34.19 | 24.77 | 46.05 | 0.368 | 0.743 | 5.18 |
| Gemini-2.5-pro | 27.59 | 28.34 | 26.72 | 22.47 | 0.254 | 0.739 | 5.77 | 21.40 | 31.16 | 29.65 | 29.30 | 40.58 | 0.348 | 0.741 | 5.63 |
| GPT-5 | 49.50 | 36.63 | 35.64 | 30.69 | 0.213 | 0.690 | 6.85 | 56.05 | 21.74 | 32.79 | 40.81 | 48.49 | 0.297 | 0.702 | 6.31 |
| RadarQA | **66.17** | 53.31 | **48.94** | **80.52** | 0.436 | **0.815** | 6.87 | 61.51 | 67.67 | 65.35 | 69.19 | 78.60 | 0.512 | **0.809** | **6.58** |
| Omni-Weather | 61.79 | **64.05** | 45.19 | 67.29 | **0.446** | 0.810 | **7.48** | **64.30** | **92.21** | **88.72** | **91.4** | **91.74** | **0.543** | 0.760 | 6.03 |

dictionreference alignment and LLM-based external judgments. To ensure fair comparison, we benchmark against the strongest available models: CasCast, DiffCast, and EarthFormer for nowcasting; WeatherGFM, UNet, and ViT for satellite-to-radar inversion; and GPT4-Score as well as the domain-specialized RadarQA for understanding. Full details of metrics and baselines are provided in Appendix A.3.

## 4.2 EXPERIMENTAL RESULTS

Currently, there exists no unified model capable of simultaneously handling both weather generation and weather understanding tasks. Existing approaches are typically specialized, such as Cascast or DiffCast for forecasting, or understanding-only models such as RadarQA for evaluation. In contrast, Omni-Weather is designed as a single framework to integrate generation and understanding. While aiming for strong quantitative performance across generation and understanding tasks, our experiments further investigate how a unified framework supports mutual gains between these tasks, reveals trade-offs between perceptual reasoning and pixel-level accuracy, and leverages both scientific and general-domain data for improved learning.

**Omni-Weather achieves superior performance in weather generation.** As shown in Table 1, Omni-Weather improves both deterministic accuracy and perceptual quality in nowcasting. Compared with single-task baselines, our model reduces CRPS by over 15% and improves LPIPS by more than 25%, while maintaining similar CSI and SSIM. On the radar inversion task, Omni-Weather consistently surpasses both specialized (i.e., WeatherGFM) and generalist (i.e., UNet and ViT) models, achieving higher CSI scores across all thresholds, with gains up to 20% at high-value levels. Furthermore, when augmented with *thinking* inference, Omni-Weather achieves clear improvements in perceptual quality, LPIPS decreases by nearly 10% while showing minor reductions on pixel-level metrics such as CSI-Mean. This highlights that explicit reasoning can enhance visual fidelity and interpretability with limited cost to deterministic accuracy.

Table 2: Training only Understanding (U), only Generation (G), or Joint (U+G). Frame / Sequence tasks are evaluated in accuracy, GPT4-Score for understanding, CSI-M, RMSE for generation.

| Setting | Understanding | | Generation | |
|---|---|---|---|---|
| | Accuracy ↑ | GPT4-score ↑ | CSI-M ↑ | RMSE ↓ |
| Und-only | 81.95 / 54.34 | 5.78 / **6.03** | - | - |
| Gen-only | - | - | 0.303 / 0.323 | 0.590 / 19.01 |
| Joint (U+G) | **86.65 / 59.58** | **7.48 / 6.03** | **0.338 / 0.347** | **0.514 / 17.11** |

Table 3: **Effect of CoT finetuning and thinking inference.** ↑ higher is better, ↓ lower is better. Training both with CoT finetuning and thinking inference achieves the best result in most metrics. Abbreviation: R.S - Radar-Score.

| CoT FT | Think Inf. | CSI-M ↑ | CRPS ↓ | R.S ↑ | LPIPS ↓ | GPT4-Score ↑ |
|---|---|---|---|---|---|---|
| ✓ | ✗ | **0.347** | **0.023** | 2.423 | 0.182 | - |
| ✗ | ✓ | 0.237 | 0.042 | 2.032 | 0.213 | 4.21 |
| ✓ | ✓ | 0.335 | **0.023** | **2.657** | **0.163** | **7.82** |

**Omni-Weather delivers strong results in weather understanding.** Table 1 also presents results for weather understanding tasks. Closed-source LLMs fail to adapt to this task, often achieving accuracies below 30%. While RadarQA serves as a competitive benchmark, Omni-Weather surpasses it: on radar image understanding, accuracy on key attributes (e.g., *Miss* and *False Alarm*) exceeds RadarQA by 20–25 points, and on radar sequence understanding, *Dynamic Consistency* improves by over 10 points with a 5% overall gain. These results highlight that Omni-Weather attains strong capability in understanding both weather sequences and single frames data.

**Omni-Weather demonstrates versatile qualitative performance across tasks.** Figure 5 illustrates Omni-Weathers outputs across both generation and understanding tasks. In the radar inversion task, Omni-Weather generates VIL fields with richer high-value structures and reasoning traces linking satellite cues to radar responses. In the radar nowcasting task, forecasts exhibit fine-grained storm details with improved spatial coherence, and the think traces offer interpretable accounts of storm evolution. For radar understanding tasks, Omni-Weather delivers expert-like outputs, combining attribute-level ratings with detailed textual evaluations that provide domain-specific insights.

## 4.3 ABLATION STUDIES AND ANALYSIS

**Impact of joint training on generation and understanding.** To examine whether generation and understanding tasks can benefit each other, we performed supervised fine-tuning on Bagel-7B-MoT using only understanding data (U-only), only generation data (G-only), or both jointly (U+G), and evaluated on 200 randomly sampled validation examples per task. As shown in Table 2, joint training improves performance across both understanding and generation: understanding achieves higher overall scores and better consistency, while generation gains in both accuracy and perceptual quality. These results indicate that unified training enables the model to both learn and interpret weather data more effectively, with generation and understanding tasks mutually enhancing each other.

**Effectiveness of Mixed Scientific and General Data.** We conducted an additional experiment by finetuning Bagel-7B-MoT on SEVIR alone versus SEVIR combined with 20k samples from the general metaquery Pan et al. (2025) dataset. As shown in Figure 6, the inclusion of general data consistently improves model performance, particularly in deterministic metrics and perceptual quality indicators. These findings suggest that while scientific data anchors domain-specific fidelity, general data provides auxiliary coverage of diverse patterns, enabling the model to learn more robust cross-modal information for better representations.

**Perceptual Trade-offs in Reasoning.** We investigate the impact of CoT-annotated supervision and reasoning-based inference on radar nowcasting, evaluating 200 carefully sampled test cases. As shown in Table 3, reasoning introduces a clear trade-off: richer prompts yield more detailed generations and improvements in perceptual image metrics (e.g., LPIPS and Radar-Score), while pixel-level measures such as CSI moderately decline. Beyond images, we also evaluate the generated textual explanations with a GPT4-Score, which assigns higher scores to CoT-enhanced outputs, confirming gains in interpretability. A case study in Figure 7 further illustrates this effect, where reasoning produces sharper storm structures and more coherent temporal evolution despite lower CSI, suggesting that it prioritizes semantic and structural fidelity over pixel-wise alignment. Qualitative comparisons of reasoning content are provided in Appendix A.5.

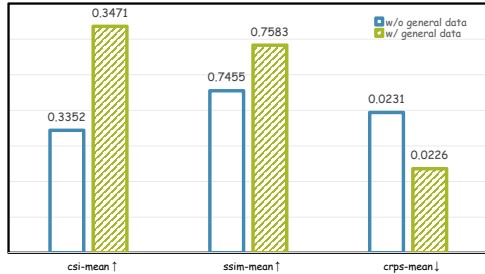

Figure 6: **Effect of mixed data.**

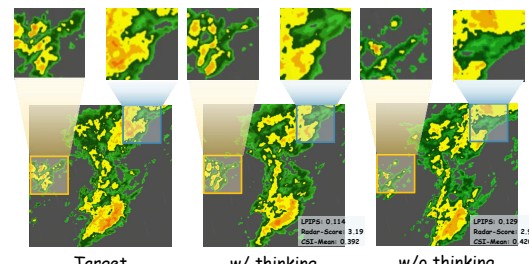

Figure 7: **Case study with thinking.**

## 5 CONCLUSION

We introduce Omni-Weather, a unified foundation model for weather generation and understanding. Through a shared backbone, it supports both generation and understanding tasks. This design allows the model to surpass task-specific baselines and enables reasoning analysis. The reasoning ability further enhances interpretability and improves the perceptual quality of radar sequences, highlighting the potential of Omni-Weather as a generalist foundation model for future weather applications. Moreover, by coupling prediction with diagnostic explanations, Omni-Weather offers a practical step toward event-level weather intelligence that is easier to audit and debug, rather than optimizing for metrics alone. We expect this unified formulation to facilitate broader multi-task transfer and more scalable evaluation protocols across heterogeneous meteorological modalities.

**Limitations.** First, Omni-Weather cannot yet adapt to general-domain VAEs. Second, broader validation across diverse weather tasks, such as medium-range forecasting and typhoon track prediction, remains necessary. Third, reasoning traces may be imperfect and require stronger faithfulness guarantees (e.g., tighter alignment between textual rationales and generated/predicted fields). Addressing these limitations will be crucial for advancing foundation models toward more robust and universally applicable weather intelligence.

## ETHICS AND REPRODUCIBILITY STATEMENT

This work develops a unified model for weather generation and understanding, aiming to improve interpretability and perceptual quality in meteorological forecasting. The datasets used (e.g., SEVIR, RadarQA) are publicly available, and all experiments follow their licenses without involving sensitive information. We acknowledge that unified models may be misused without expert oversight; thus, our method is intended for research purposes and should be complemented by professional interpretation. To support reproducibility, we provide detailed descriptions of tasks, model design, training objectives, and evaluation protocols, and we release a Chain-of-Thought dataset for causal reasoning at `https://anonymous.4open.science/r/cot-data-E3D4/` along with its construction code. Upon acceptance, we will further release the complete codebase, model checkpoints, and data-processing scripts to ensure transparency and verifiability.

## ACKNOWLEDGMENTS

This work is supported by Shanghai Artificial Intelligence Laboratory.

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

# A APPENDIX

## A.1 OVERVIEW

This Appendix is structured as follows:

- Sec. A.2: Details of task paradigm and dataset.
- Sec. A.3: Details of evaluation metrics.
- Sec. A.4: Data construction of Chain-of-thought dataset.
- Sec. A.5 : Details about Chain-of-though reasoning.
- Sec. A.6: More experimental results of Omni-Weather.
- Sec. A.7: Qualitative results of Omni-Weather.
- Sec. A.8: Usage of LLM.

## A.2 MORE DETAILS ABOUT DATASETS AND TASK

### A.2.1 DATA DETAIL

**SEVIR.** The Storm EVent ImagRy (SEVIR) dataset is a large-scale collection of temporally aligned weather observations covering the continental United States. It integrates multiple sensing modalities, including visible and infrared satellite imagery, lightning event records, and mosaics of Vertically Integrated Liquid (VIL) derived from NEXRAD radar. In this work, we focus on the radar-based VIL product, which provides a spatio-temporal representation of convective storm structures. SEVIR contains over 20,000 storm events sampled between 2017 and 2020, each spanning a 4-hour window at 5-minute resolution and covering approximately $384\,\mathrm{km} \times 384\,\mathrm{km}$ regions. To support short-range forecasting tasks, sequences are typically arranged as input–output pairs, where a set of observed frames is used to predict future VIL evolution. The images are normalized to the range $[0, 255]$ and evaluated against threshold-based metrics (e.g., CSI, HSS) following established protocols. This combination of multi-sensor coverage, temporal alignment, and standardized evaluation makes SEVIR a widely adopted benchmark for data-driven weather prediction.

**RadarQA Dataset.** RQA-70K from RadarQA is a large-scale forecast quality analysis dataset encompassing four tasks: frame rating, frame assessment, sequence rating, and sequence assessment. RQA-70K is constructed through a combination of human annotation and automated labeling. By integrating traditional forecasting metrics with expert knowledge, the dataset provides a comprehensive benchmark for the assessment of weather radar forecasting.

### A.2.2 TASK DETAIL

**Weather Generation.** For radar nowcasting, the model is trained to predict the short-term spatio-temporal evolution of precipitation. Specifically, we use sequences of 10 observed VIL frames as input and require the model to generate the subsequent 12 frames. All frames are preprocessed to a spatial resolution of $256 \times 256$, which balances coverage of mesoscale convective features with computational efficiency. This setting follows standard short-range nowcasting protocols but is tailored to emphasize fine-scale storm structures, ensuring that the model learns both spatial coherence and temporal continuity in convective system development.

**Weather Understanding.** Weather understanding tasks require the model to produce natural-language descriptions and evaluations based on radar observations or forecast sequences. The input can be either a single VIL frame or a sequence of frames, and the output is a structured report covering key meteorological aspects such as storm morphology, intensity, temporal evolution, and forecast quality (e.g., hits, misses, or false alarms). Following the RadarQA benchmark, model responses are rated along multiple dimensions using a four-level ordinal scale (*fair*, *poor*, *good*, *great*), which are mapped to numerical values 14. Scores are then averaged across dimensions to obtain the *Radar Score*, providing a comprehensive indicator of diagnostic quality. This task formulation bridges language modeling with domain-specific evaluation, enabling models not only to assess physical forecasts but also to generate expert-like reasoning aligned with meteorological practice.

A.3 EVALUATION PROTOCOLS AND METRICS

To enable a more comprehensive and accurate assessment, we employ a diverse set of evaluation metrics across different tasks. The detailed definitions of these metrics are provided below.

**CSI**. Critical Success Index (CSI) is widely used in the evaluation for weather forecasting tasks. Formally, it is defined as:

$$CSI = \frac{TP}{TP + FN + FP} \tag{A.1}$$

where $TP$, $FP$, and $FN$ denote the number of true positives, false positives, and false negatives, respectively. Following Cascast Gong et al. (2024), we apply thresholds at 16, 74, 133, 160, 181, and 219 to evaluate model performance across different VIL intensity ranges.

**SSIM** Wang et al. (2004). Structural Similarity Index Measure (SSIM) is a perceptual metric that quantifies the similarity between two images by comparing their contrast, luminance, and structure. Formally, it is defined as:

$$SSIM(x, y) = \frac{(2\mu_x\mu_y + C_1)(2\sigma_{xy} + C_2)}{(\mu_x^2 + \mu_y^2 + C_1)(\sigma_x^2 + \sigma_y^2 + C_2)} \tag{A.2}$$

where $\mu_x$ and $\mu_y$ are the means of $x$ and $y$, $\sigma_x^2$ and $\sigma_y^2$ are the variances, $\sigma_{xy}$ is the covariance, and $C_1$, $C_2$ are small constants to stabilize the division. Higher SSIM values indicate stronger similarity between the prediction and observation.

**CRPS**. Continuous Ranked Probability Score (CRPS) evaluates the accuracy of probabilistic forecasts by comparing Cumulative Distribution Function (CDF) with observation $x$. Formally, it is defined as:

$$CRPS(F, x) = \int_{-\infty}^{+\infty} \big(F(y) - \mathbf{1}\{y \geq x\}\big)^2 dy \tag{A.3}$$

where $\mathbf{1}\{y \geq x\}$ is the indicator function.

**LPIPS** Zhang et al. (2018). Learned Perceptual Image Patch Similarity (LPIPS) is a perceptual metric designed to evaluate similarity between images in a manner aligned with human judgment. LPIPS leverages neural networks to compute differences in deep feature-based representations.

**Rouge_L Lin (2004)**. Rouge_L is widely used for evaluating the quality of generated text by measuring the Longest Common Subsequence (LCS) between a candidate and reference sequence. Rouge_L accounts for sentence-level similarity, capturing both content and fluency.

**BertScore Zhang et al. (2019)**. BertScore is a learned mertic for evaluating text generation that leverages text embeddings from pretrained language models (e.g., Bert) to compute similarity between candidate and reference sentences.

**GPT4-Score**. GPT4-Score leverages the reasoning and understanding capabilities of GPT4 to assess generated outputs. Specifically, both the ground truth and the prediction are provided to GPT-4, which evaluates the prediction based on overall accuracy, content richness, and fidelity to the reference.

**Radar-Score**. RadarQA formulates radar understanding as a rating task, where model-generated diagnostic reports are assessed along multiple meteorologically relevant dimensions, such as storm morphology, intensity, temporal evolution, and forecast quality. Each dimension is rated on a four-level ordinal scale {fair, poor, good, great}, which are mapped to numerical values 1 to 4. The *Radar Score* is then obtained by averaging these ratings across all evaluated dimensions, producing a single interpretable measure that reflects the overall diagnostic quality of the models output.

A.4 DETAIL OF CHAIN-OF-THOUGHT CONSTRUCTION

To enable causal reasoning over storm dynamics in the weather domain, we carefully design a CoT data construction pipeline to generate high-quality CoT data.

**Data Preprocess**. First, we extract the raw data from the SEVIR dataset and segment each event into three pairs of 10-frame inputs and 12-frame outputs. Second, we filter out samples with limited informative content and visualize the retained frames using SEVIR's colormap.

**Attributes Annotation**. For the constructed input / output frame pairs, we perform attributes annotation by designing a structured prompt, which encompasses four components: system prompt,

attribute options, return format, and caution instructions. The prompt is then provided to a large language model to generate structured JSON outputs that serve as inputs for the subsequent CoT annotation process.

---

**System Prompt for Attribute Annotation**

You are a meteorological expert. You will be given an animated GIF containing the 10/12 LABEL frames to generate the final conclusions. Use the taxonomy below and produce a complete "analysis_label" covering all attributes, consistent with the LABEL sequence. Use only the exact option strings provided for each attribute; if an attribute is not apparent from the LABEL frames, set "choice" to "not apparent". In "rationale", use probabilistic language when evidence is weak (e.g., "likely", "possible"). Do not reference or speculate about INPUT frames.

**Orientation & mapping (do not ignore)**:
Images are rendered with 'origin='upper''. Treat the top of the image as North (N), the bottom as South (S), the left as West (W), and the right as East (E). - Up = North, Down = South, Left = West, Right = East. - Diagonals: Up-Left = Northwest, Up-Right = Northeast, Down-Left = Southwest, Down-Right = Southeast. Do not rotate or flip the images. Determine 'Main motion direction' strictly using this mapping.

**Definitions (used consistently throughout):**
- Convective cell: a contiguous region whose pixel value exceeds a reflectivity/intensity threshold (>32 on a 0255 scale; colorbar green and above). Very small isolated speckles can be ignored when they are unlikely to influence scene-level judgment. - Main convective system: the dominant connected system within the scene (the most spatially prominent/organized aggregate of cells). When multiple candidates exist, use the overall/aggregate pattern that best represents the sequence. Taxonomy (attributes and allowed options):{enum_text} Return a strict JSON object in English with the exact schema:{schema_text}

---

**Attribute Options**

"Morphology": (
    "Spatial organization pattern at the start window (or across the input window); "
    "scattered = many small isolated cells distributed broadly (low organization); "
    "banded = elongated/narrow rainband aligned along a main axis; "
    "blob-like = one or a few dominant compact clusters with larger areal extent; "
    "spiral = curved/spiraling rainbands (e.g., tropical cyclones/mesoscale vortices); "
    "layered = sheet-like, wide coverage with uniform texture and less distinct boundaries;
"
    "bow-shaped = an arced/bowing reflectivity segment (e.g., bow echo)."
),
"Max pixel level": (
    "Peak intensity represented by the maximum pixel value across the scene (0255). "
    "Use the provided bins: no significant (031), very weak (3173), weak (74132), "
    "moderate (133159), strong (160180), very strong (181218), extreme (219255)."
    " Pixel intensity values map directly to the provided colorbar."
),
"Initial position of the main convective system": (
    "Coarse location of the dominant connected system at t0 in image coordinates; "
    "Centered / N / S / W / E / four quadrants; or 'no clear main system' if dominance is ambiguous."
),
"Main motion direction": (
    "Dominant displacement of the MAIN system (aggregate of cells above threshold >32) across frames. "
    "If multiple clusters move differently or displacement is unclear, choose 'no obvious motion'."
)

```
    ),
    "Main motion speed": (
        "Qualitative speed class based on normalized displacement per frame relative to image
size: "
        "near-stationary / slow / moderate / fast / very fast."
    ),
    "Rotation center": (
        "Approximate sector of the rotation center if clear cyclonic/anticyclonic rotation is
present (N/NE/E/SE/S/SW/W/NW); "
        "otherwise 'no rotation' or 'location uncertain'."
    ),
    "Change in number of convective cells": (
        "Trend in the count of distinct convective cells. "
        "Ignore tiny isolated speckles when appropriate."
    ),
    "Morphological evolution of the main system": (
        "Primary structural change of the MAIN system over time: elongation, shrinkage, ex-
pansion, "
        "merging, splitting, dissipation, generation."
    ),
    "Intensity evolution": (
        "Overall trend of reflectivity/brightness of the MAIN system: strengthening / weaken-
ing / roughly unchanged."
    ),
    "Areal coverage evolution": (
        "Trend in the areal extent of above-threshold pixels (>32): "
        "expanding / roughly unchanged / rapidly shrinking / expand then shrink / shrink then
gradually expand."
    ),
    "Organization evolution": (
        "Change in connectedness/ordering of the system: becoming connected (fragmented
more coherent), becoming fragmented (coherent  more broken), "
        "connected then gradually weakening (coherent but loosening/fading), fragmented then
becoming connected (consolidation), no obvious change."
    ),
```

## Return Format for Attribute Annotation

The output must be valid JSON (no markdown fences), with the following keys:
```
{
    "sample_id": str,
    "analysis_input": {
        "annotations": {
            "Morphology": {"choice": str, "rationale": str},
            "Max pixel level": {"choice": str, "rationale": str},
            "Initial position of the main convective system": {"choice": str, "rationale":
str},
            "Main motion direction": {"choice": str, "rationale": str},
            "Main motion speed": {"choice": str, "rationale": str},
            "Rotation center": {"choice": str, "rationale": str},
            "Change in number of convective cells": {"choice": str, "rationale": str},
            "Morphological evolution of the main system": {"choice": str, "rationale":
str},
            "Intensity evolution": {"choice": str, "rationale": str},
            "Areal coverage evolution": {"choice": str, "rationale": str},
            "Organization evolution": {"choice": str, "rationale": str}
        },
```

```
        "global_rationale": str
    }
}
```

---

**Additional Rules for Attribute Annotation**

Additional rules:
    - Consider only INPUT frames (t0..t9); do not mention LABEL frames.
    - Cover all attributes above.
    - For each attribute, 'choice' must be one of the listed options verbatim, or the literal string "not apparent" when evidence is insufficient.
    - Keep explanations concise, using terms like "likely" / "possible" when appropriate.
    - Orientation & mapping (do not ignore): Images are rendered with 'origin='upper''. Treat the top of the image as North (N), the bottom as South (S), the left as West (W), and the right as East (E). For diagonals: up-left=Northwest, up-right=Northeast, down-left=Southwest, down-right=Southeast. Do not rotate or flip the images.
    - Consistency check for "Main motion direction": Verify that the apparent displacement on the image (screen coordinates) matches the above mapping before choosing. If uncertain, choose "no obvious motion".
    - Definition of "Main motion direction": Determine motion using the contiguous regions where pixel value > 31 (i.e., colorbar green and above). Track the overall trajectory of these regions from t0 to t9 (or t10 to t11 for LABEL). If multiple such regions exist, report their general/aggregate motion direction. If regions are too fragmented or exhibit multiple inconsistent directions, set the direction to "no obvious motion".

---

**CoT Annotation**. After obtaining the annotated attribute data, the large language model is prompted to generate outputs in a predefined sequence: task instruction, Temporal causal factor, perceptual causal factor, direct outcomes, and deep outcomes. The system prompt, cautions, return format, and instruction for each step are detailed as follows.

---

**System Prompt for CoT Annotation**

You are a senior meteorological nowcasting expert building a high-quality CoT dataset. Write in precise, fluent ENGLISH. For every step, produce flowing sentences only. Do NOT use bullet points, numbered lists, dashes, or tables. Use the exact taxonomy key names verbatim when referring to keywords; never rename or invent new keys. Init thought must rely ONLY on INPUT annotations; Summary thought must rely ONLY on LABEL annotations. Keep the reasoning scientific and case-specific.

---

**Instructions and Examples for each Step**

INIT THOUGHT (INPUT-based; do not use LABEL):
**Step 1 Task instruction:**
Explain in 23 sentences that this is a short-range precipitation nowcasting task based on the last 10 VIL frames, and the goal is to forecast the next 12 frames.
**Step 2 Temporal causal factor extraction (merged with temporal description):**
Write a single descriptive paragraph that uses ONLY the temporal causal factor keywords. Explicitly restate the choices for "Main motion direction", "Main motion speed", and "Rotation center" as observed from t0t9, and keep the prose strictly descriptive. Do NOT include meta-summaries such as this establishes a premise, avoid invented terminology, mechanisms, or evaluative language, and do not reference perceptual or outcome keywords. Use present/immediate past tense and clear compass terms.
Example (style only): "Across t0t9, 'Main motion direction' is east, 'Main motion speed' is slow, and 'Rotation center' is no rotation, so the motion is a slow eastward translation without rotational signatures."
**Step 3 Perceptual causal factor extraction (merged with spatial description):**

Write a single descriptive paragraph that uses ONLY the perceptual causal factor keywords. Explicitly restate the choices for "Morphology", "Max pixel level", and "Initial position of the main convective system" as observed from t0t9, and keep the prose strictly descriptive. Do NOT include meta-summaries such as this defines a baseline, avoid mechanisms or evaluative language, and do not reference temporal or outcome keywords. Use the present/immediate past tense.

Example (style only): "The 'Morphology' is bloblike, the 'Max pixel level' is strong, and the 'Initial position of the main convective system' is centered, yielding compact highintensity echoes near the center across t0t9."

**Step 4 Direct outcomes (1stTier Outcome):**

Analyze ONLY the direct outcome keyword "Intensity evolution". Write exactly ONE concise sentence that delivers the conclusion for "Intensity evolution", justified solely by the temporal and perceptual paragraphs (Steps 23). The sentence must explicitly name "Intensity evolution". Do NOT reference deep outcomes, mechanisms, grids, coordinates, or any terms not present in the keyword taxonomy. Avoid generic phrases like "structural pressure" or "integration forces".

Example (style only): "For 'Intensity evolution', the strong yet steady baseline together with slow translation and no rotation supports an assessment that intensity remains roughly unchanged."

**Step 5 Deep outcomes (2ndTier Structural Outcome):**

Analyze ONLY the deep outcome keywords "Areal coverage evolution" and "Organization evolution". Write exactly TWO sentences in a single flowing paragraph: one sentence for "Areal coverage evolution" and one for "Organization evolution". Each sentence must deliver a concise conclusion justified by the temporal and perceptual paragraphs (Steps 23) and the direct outcome (Step 4), and must explicitly name the corresponding deep outcome keyword. Do NOT introduce mechanisms here; do NOT add invented terminology or extra commentary.

Example (style only): "'Areal coverage evolution' remains stable under slow translation and steady intensity; 'Organization evolution' shows no obvious change because the bloblike structure and absence of rotation provide no pathway to linearization or fragmentation."

**Final summary paragraph:**

After completing Steps 15, write a short paragraph (24 sentences) that integrates the INPUT-based reasoning across all steps into a coherent training target. Keep it as flowing prose with no lists. Place it under "init_thought.summary".

**QUality Control**. After obtaining the annotated CoT data, we perform a three-step quality control: Structure Check, Causal Alignment, and Terminology. Data that pass all steps are retained for training and included in the final CoT dataset.

Finally, we obtained 4,000 CoT annotations for radar nowcasting and 4,000 CoT annotations for radar inversion, which together form our CoT dataset for generation tasks.

## A.5 MORE DETAIL ABOUT REASONING

We provide qualitative comparisons between Omni-Weather with and without CoT finetuning on both radar inversion and radar nowcasting tasks. Figures 8 and 9 present three representative cases for radar inversion and radar nowcasting. As shown, the CoT-finetuned model produces reasoning traces that not only accompany the generated radar fields with higher perceptual quality, but also deliver interpretable textual explanations grounded in storm dynamics. In contrast, the non-CoT model tends to generate thinking outputs that resemble post-hoc quality evaluations rather than causal reasoning, lacking direct connection to the generation process itself. This highlights that CoT supervision guides the model toward producing reasoning that is both explanatory and predictive, effectively bridging image generation with meteorological interpretation.

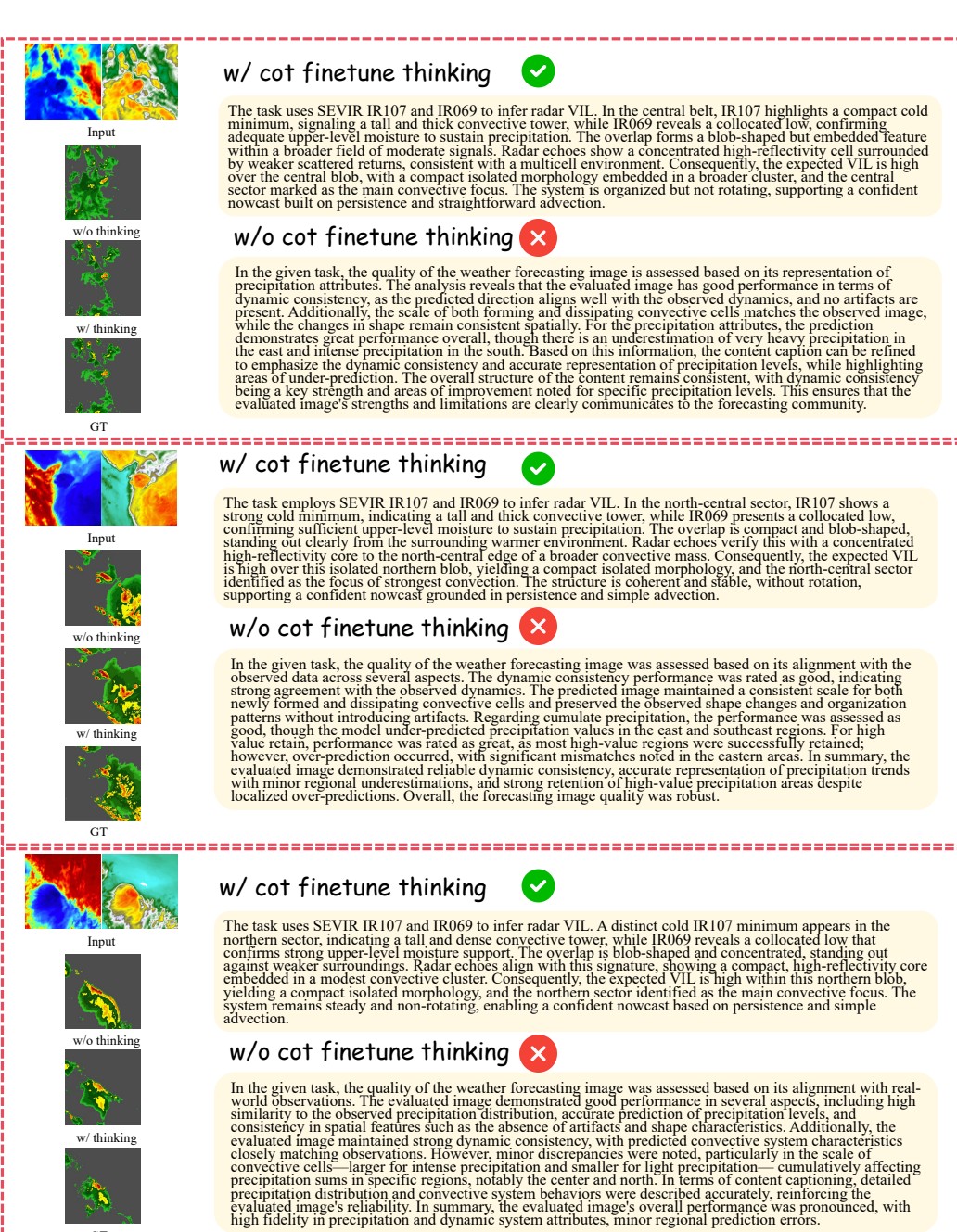

Figure 8: **Radar Inversion Thinking Comparison**.

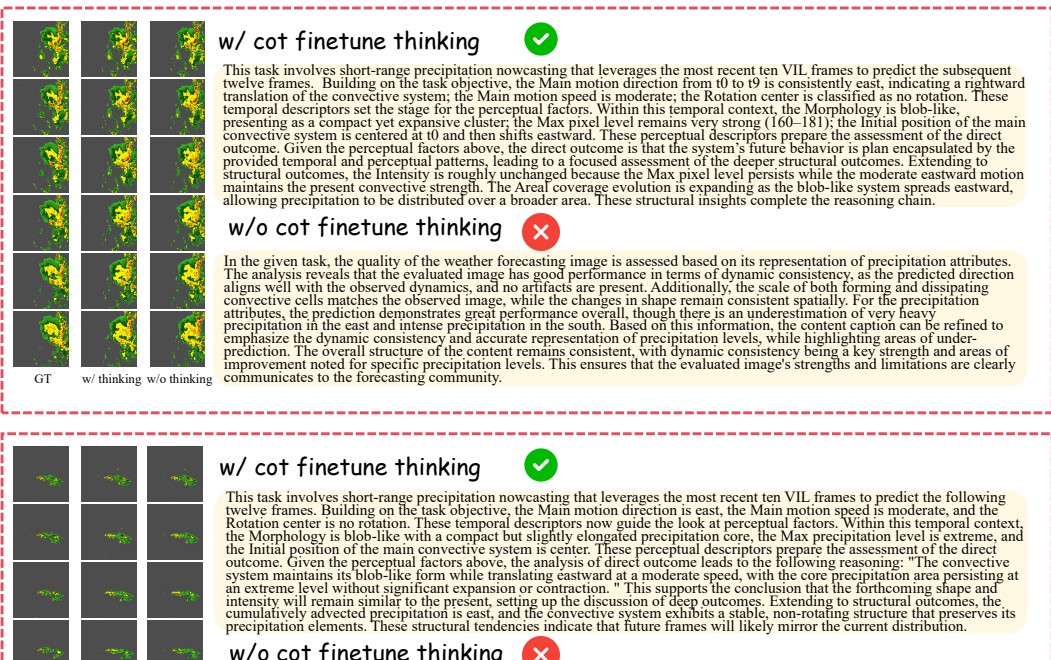

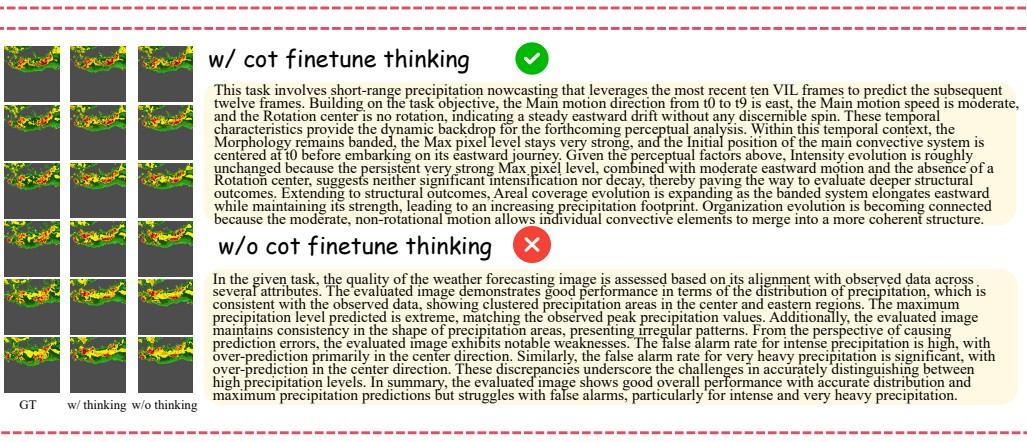

Figure 9: **Radar Nowcasting Thinking Comparison**.

A.6    MORE EXPERIMENT RESULTS

All ablations in this section are conducted on the **SEVIR** test set with 200 sequences under the *now-casting* task. We report *CSI-mean*, *CSI-pool4-mean*, *CSI-pool16-mean*, *SSIM*, and *PSNR*. Higher values are better for all metrics.

**Radar Sequence Encoder vs. VAE Encoder**    To validate the effectiveness of the proposed radar sequence encoder, we compare it against a vanilla VAE encoder. As shown in Table 4, the radar sequence encoder achieves consistent improvements across all CSI metrics, SSIM, and PSNR, demonstrating that modeling temporal radar sequences brings substantial gains.

Table 4: Comparison of radar sequence encoder and VAE encoder.

|  | CSI-mean | CSI-pool4-mean | CSI-pool16-mean | SSIM | PSNR |
|---|---|---|---|---|---|
| VAE encoder | 0.2358 | 0.2912 | 0.4356 | 0.7528 | 21.42 |
| Radar sequence encoder | **0.3471** | **0.4003** | **0.5390** | **0.7621** | **23.22** |

**Mixing General and Weather Data**    We further explore the impact of mixing general-purpose data ("gen") and weather radar data ("weather"). Here, the notation "1gen30%+weather70%" means that one general dataset with 30% proportion is combined with 70% weather data, while "2gen30%+weather70%" means two general datasets each contributing 30% combined with 70% weather data. As reported in Table 5, using a single general dataset at 30% ratio achieves the best balance and outperforms other settings.

Table 5: Results under different generation data mixing strategies.

|  | CSI-mean | CSI-pool4-mean | CSI-pool16-mean | SSIM | PSNR |
|---|---|---|---|---|---|
| 1gen30%+weather70% | **0.2501** | **0.2994** | **0.4261** | **0.6866** | **19.67** |
| 1gen70%+weather30% | 0.1386 | 0.1726 | 0.2859 | 0.6187 | 16.66 |
| 1gen50%+weather50% | 0.2478 | 0.2956 | 0.4174 | 0.6823 | 19.15 |
| 2gen30%+weather70% | 0.1091 | 0.1345 | 0.2274 | 0.6013 | 16.64 |

**CFG Setting**    Lastly, we examine the choice of the classifier-free guidance (CFG) parameter. As shown in Table 6, setting CFG= 2 yields better overall performance compared with CFG= 1, especially on CSI metrics, SSIM, and PSNR. Therefore, we adopt CFG= 2 as the default configuration.

Table 6: Ablation on CFG settings.

|  | CSI-mean | CSI-pool4-mean | CSI-pool16-mean | SSIM | PSNR |
|---|---|---|---|---|---|
| CFG=2 | **0.2501** | **0.2994** | **0.4261** | **0.6866** | **19.67** |
| CFG=1 | 0.1824 | 0.2208 | 0.3305 | 0.5123 | 17.33 |

A.7    MORE QUALITATIVE RESULTS

We provide additional qualitative examples of Omni-Weather on radar nowcasting, radar inversion, and radar understanding tasks. Figures 10–12 illustrate diverse cases, complementing the quantitative results in the main paper and highlighting the models ability to capture storm dynamics and generate interpretable assessments.

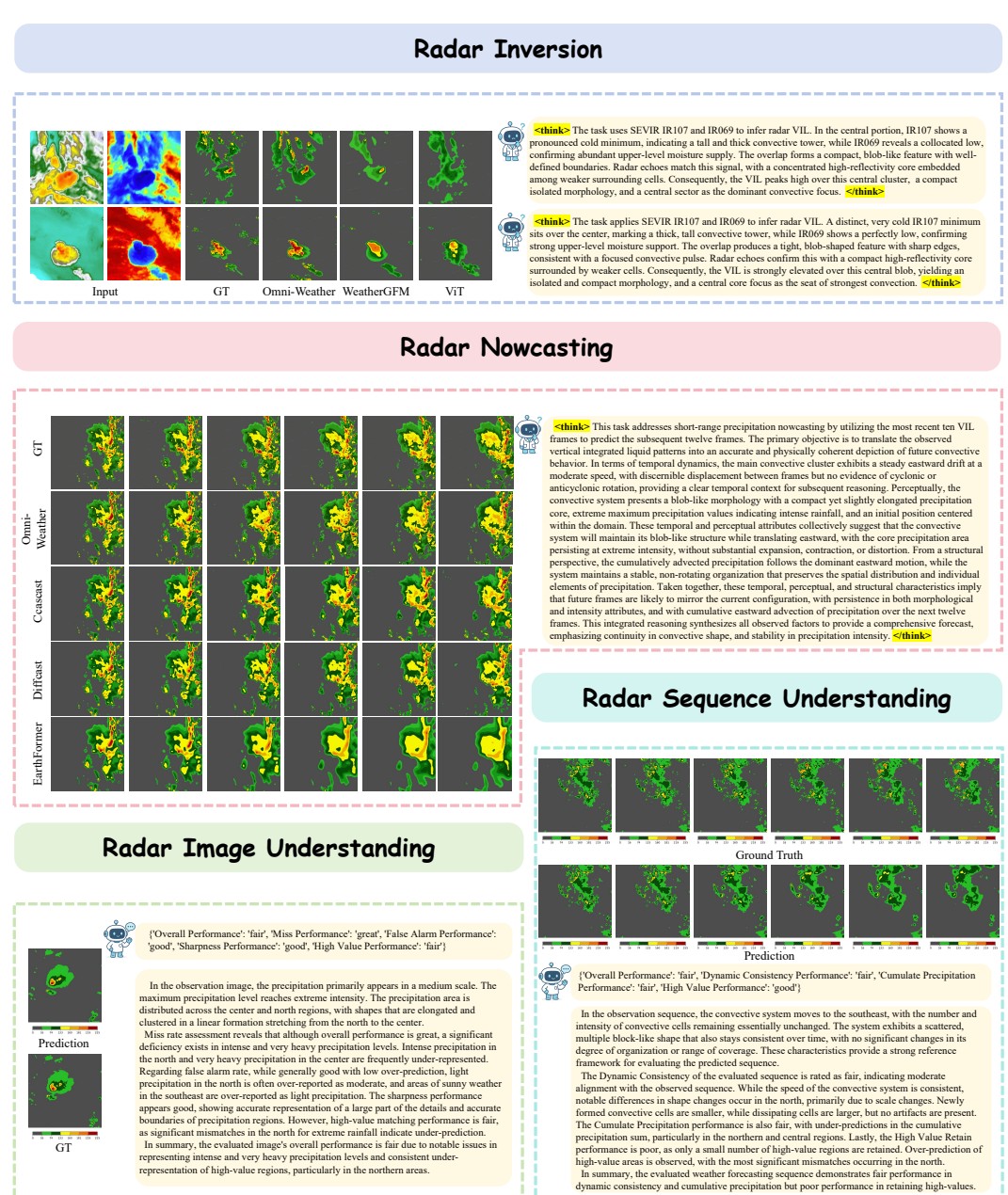

Figure 10: **Qualitative Result 1**.

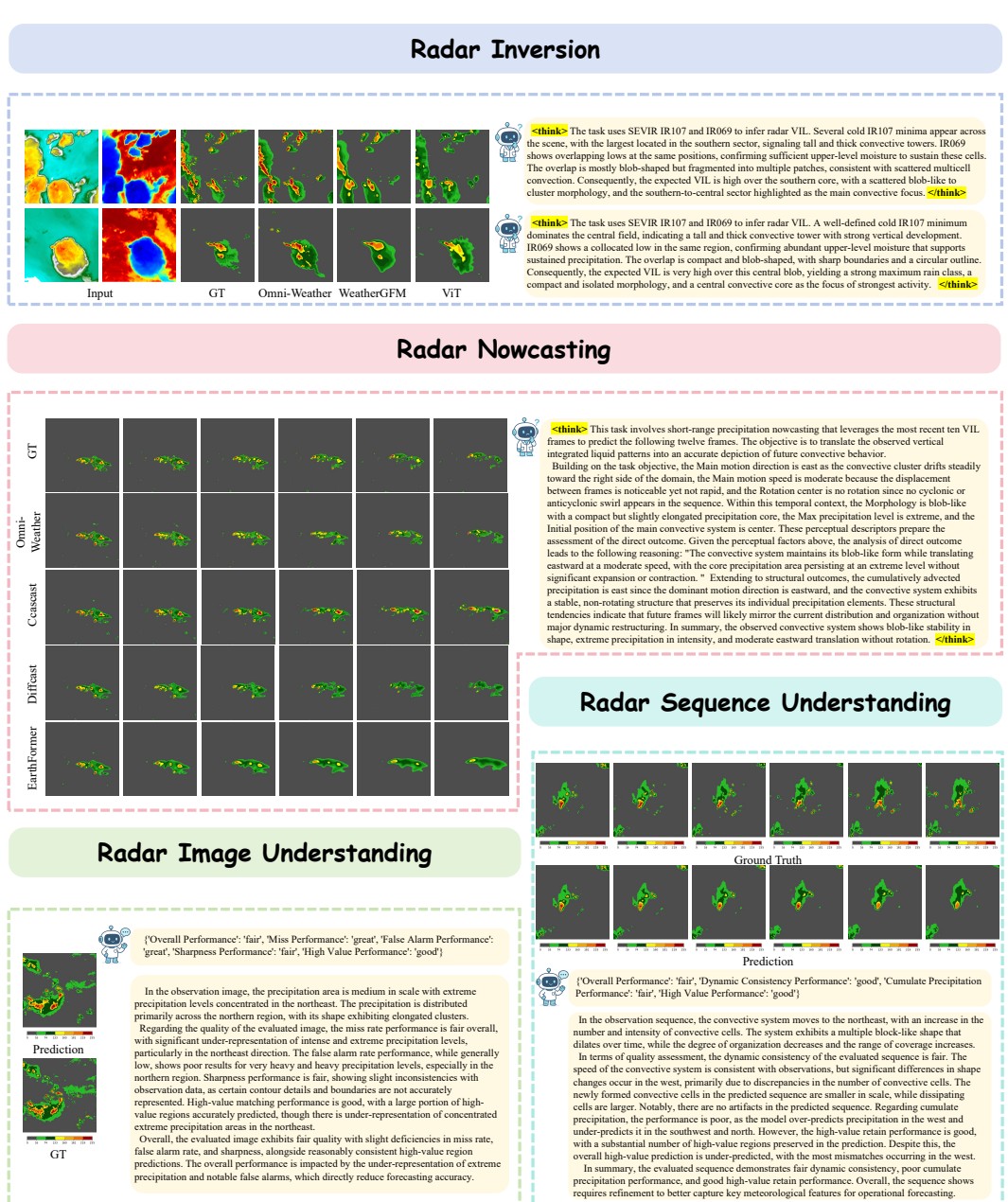

Figure 11: **Qualitative Result 2**.

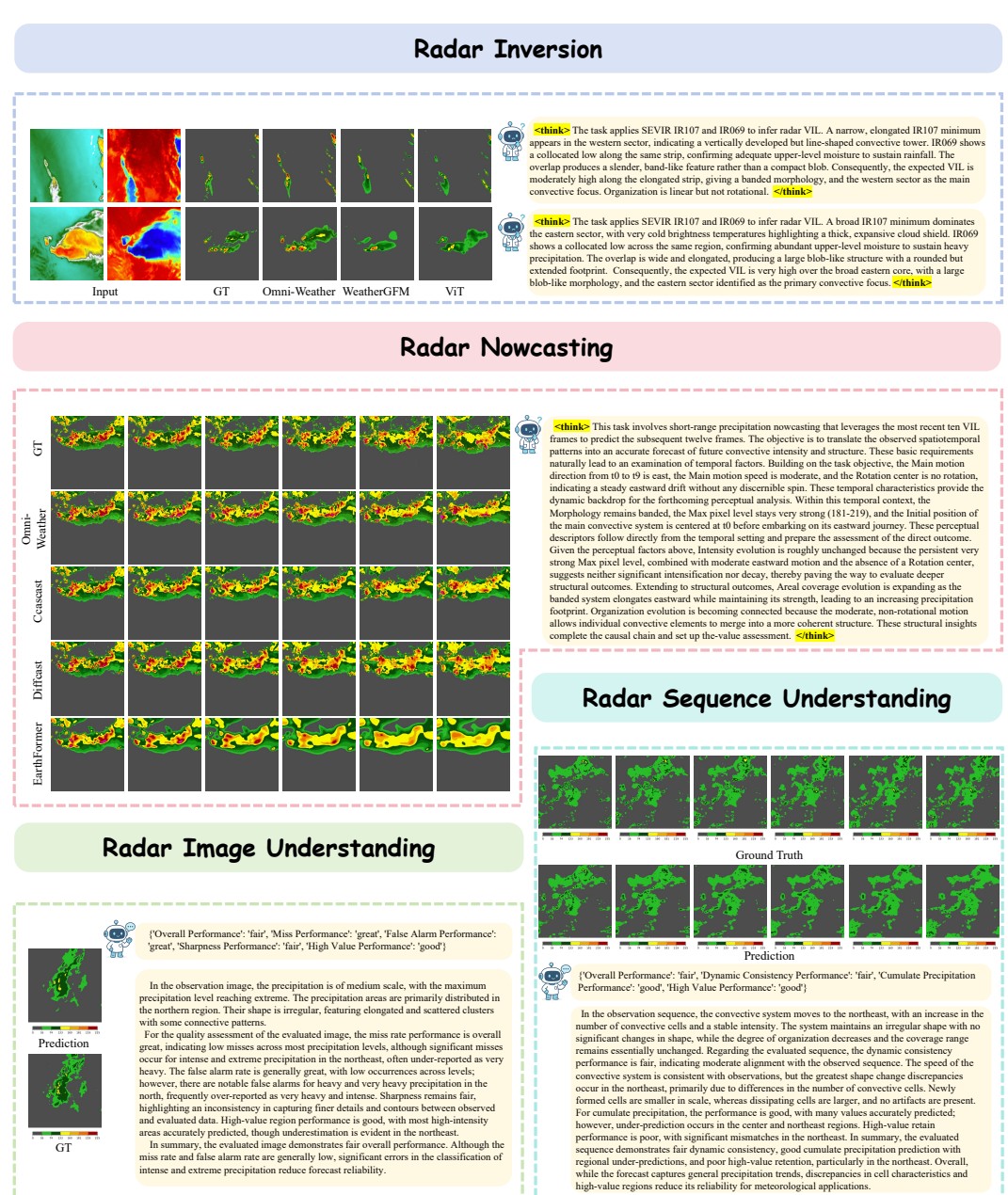

Figure 12: **Qualitative Result 3**.

## A.8 USAGE OF LLM

Large language models were employed as an auxiliary tool to support manuscript preparation, including grammar checking, sentence refinement, and clarification of technical descriptions. All AI-suggested text was carefully reviewed and revised by the authors to ensure accuracy, clarity, and scientific integrity.

