# OpenReview forum: "Omni-Weather: A Unified Multimodal Model for Weather Radar Understanding and Generation"
_ICLR.cc/2026/Conference — ICLR 2026 Poster_

### Official Review · Reviewer_3h1v · 2025-10-30

**Soundness:** 3
**Presentation:** 3
**Contribution:** 3
**Rating:** 6
**Confidence:** 3

**Summary:**

This paper presents Omni-Weather, a multimodal foundation model that unifies weather generation and understanding within a single architecture. Unlike existing models that separately address forecasting or diagnostic reasoning, Omni-Weather integrates radar and text modalities through a shared self-attention backbone and a Chain-of-Thought (CoT) dataset to enable causal reasoning in weather modeling. The model achieves state-of-the-art results on both weather generation (e.g., nowcasting, radar inversion) and understanding (e.g., RadarQA tasks), demonstrating that generative and interpretive capabilities can reinforce each other.

The contributions are:
1. Introduction of the first unified multimodal foundation model for weather that jointly handles generation (forecasting, inversion) and understanding (diagnostic reasoning, QA) tasks within a single framework.
2. Construction of a weather-specific Chain-of-Thought (CoT) dataset for causal reasoning in generation, improving interpretability and perceptual quality of outputs.
3. Empirical results showing Omni-Weather surpasses strong baselines (e.g., CasCast, DiffCast, WeatherGFM, RadarQA) in both pixel-level and perceptual metrics, with reasoning further enhancing visual fidelity and explainability.

**Strengths:**

1. The paper introduces the first unified multimodal foundation model for weather generation and understanding, representing a novel and impactful problem formulation.
2. The Chain-of-Thought dataset for causal reasoning in weather generation is promising, enabling interpretable forecasting and bridging the gap between prediction and explanation.
4. The experiments are comprehensive, covering both pixel-level and perceptual evaluations with clear comparisons to strong baselines.
5. The paper is well-written and clearly structured.
6. The demonstrated mutual benefit between generation and understanding tasks highlights significant scientific insight with implications for broader multimodal foundation model research.

**Weaknesses:**

1. The claim of a “foundation model for weather” seems overstated, as the model’s scope is limited to a single variable (radar VIL precipitation) rather than encompassing multiple atmospheric variables such as temperature, pressure and wind.
2. The proposed model only addresses short-range nowcasting (approximately one hour ahead) and is restricted to the SEVIR dataset covering the continental US, limiting its generalization and global applicability.
3. The Chain-of-Thought (CoT) dataset used for training is entirely LLM-generated, with no human expert validation or meteorological review to ensure physical correctness, as GPT-series models are not fine-tuned as meteorologist experts.
4. The CoT generation pipeline relies on GPT-4o for attribute annotation and GPT-o3 for reasoning synthesis, producing synthetic causal narratives that may not reflect authentic meteorological reasoning. An ablation or qualitative comparison between LLM-generated CoT reasoning and human meteorologist-written reasoning would clear the confound of whether the improvements stem from genuine interpretability or stylistic mimicry of GPT.

**Questions:**

Please refer to weaknesses.

---

> ### Author Response · Authors · 2025-11-23
> **Answer of W1&2 To Reviewer 3h1v**
>
> Dear Reviewer 3h1v,
>
> We appreciate the reviewer’s constructive and detailed feedback. We try to address the concerns and questions below.
>
> **W1&2: Scope and generalization of Omni-Weather beyond single-variable SEVIR radar**
>
> **A1&2:**
> We thank the reviewer for the reminder about the scope implied by calling Omni-Weather a “weather foundation model.” We agree that the current training and evaluation are concentrated on a **single physical variable** (radar VIL) and three SEVIR-based tasks: short-range radar nowcasting, satellite-to-radar inversion, and RadarQA-style understanding. As a result, the wording “weather foundation model” can indeed be misread as if the model already covered a much broader set of atmospheric variables (e.g., temperature, wind, pressure, humidity).
>
> To avoid over-claiming, we have **revised the terminology throughout the paper**. Concretely, we (i) change the title to:
>
> *“Omni-Weather: A Unified Multimodal Model for Weather Radar Understanding and Generation”*
>
> In future-ready version. and (ii) soften the use of “foundation model” in the abstract and introduction, explicitly stating that the present work **focuses on the radar modality** under the SEVIR setting rather than claiming coverage of all weather variables.
>
> In addition, we now make the model’s applicability **explicit in the Limitations section**: Omni-Weather in its current form is restricted by SEVIR’s single-variable radar data, and we do *not* claim to already handle a full suite of atmospheric variables. To nevertheless show that the framework is not intrinsically tied to SEVIR radar VIL, we add **preliminary ERA5 extension experiments**, where we augment training with multi-lead forecasts of 2 m temperature (T2m) on the WeatherBench setup and compare against IFS, ClimaX, and WeatherGFM.
>
> In this experiment:
>
> * **ClimaX** follows the original protocol and treats each forecast lead time as an independent task, fine-tuning a separate model per lead.
> * **Omni-Weather**, in contrast, keeps a **single unified model** that jointly handles SEVIR radar generation/understanding and ERA5 T2m multi-lead forecasting, without splitting by lead time or introducing extra per-lead fine-tuning.
>
> The table below summarizes RMSE and ACC for T2m:
>
> | Lead Time (h) | RMSE IFS | RMSE ClimaX | RMSE WeatherGFM | **RMSE Omni-Weather (ours)** | ACC IFS | ACC ClimaX | ACC WeatherGFM | **ACC Omni-Weather (ours)** |
> | ------------: | -------: | ----------: | --------------: | ---------------------------: | ------: | ---------: | -------------: | --------------------------: |
> |             6 |     **0.97** |        1.11 |            1.08 |                     1.07 |    **0.99** |       0.98 |           0.98 |                    0.98 |
> |            24 |     **1.02** |        1.19 |            1.23 |                     1.21 |    **0.99** |       0.97 |           0.97 |                    **0.97** |
> |            72 |     **1.30** |        1.47 |            1.56 |                     1.48 |    **0.98** |       0.96 |           0.96 |                    0.97 |
> |           120 |     1.71 |        1.83 |            1.68 |                     **1.63** |    **0.96** |       0.94 |           0.95 |                    **0.96** |
> |           168 |     2.23 |        2.17 |            1.76 |                     **1.70** |    0.93 |       0.91 |           0.94 |                    **0.95** |
>
> We observe that at **72 h, 120 h, and 168 h**, Omni-Weather achieves better RMSE and ACC than WeatherGFM, clearly outperforms ClimaX at 120 h and 168 h, and reaches or slightly surpasses IFS; at shorter leads (6 h, 24 h) our performance is comparable to WeatherGFM and ClimaX. These results do *not* claim a full global, multi-variable foundation model, but they do indicate that the **same unified architecture** can extend beyond SEVIR radar VIL to additional variables and longer forecast ranges, supporting the view of Omni-Weather as a generalizable *framework* rather than a SEVIR-only model.

---

> ### Author Response · Authors · 2025-11-23
> **Answer of W3 To Reviewer 3h1v**
>
> > W3: Reliability of LLM-generated CoT data without full expert meteorological annotation
>
> **A3:**
> We agree this is an important point, meteorologists were involved at three stages—schema/QC design, batch-level spot checks during construction, and a post-hoc gold-standard evaluation—on top of a strict automated QC pipeline. Below we summarize expert involvement and annotation reliability.
>
> 1. **Expert-designed CoT schema and QC rules before large-scale generation.**
>    Before generating data at scale, members of the author team with meteorological background manually analyzed a representative set of SEVIR cases and, based on operational practice, identified the key elements that must be covered in short-range convective explanations. From this process, they derived:
>
>    * a **four-layer causal structure** (temporal/motion factors, perceptual factors, first-level outcome inference, and higher-level structural outcome), and
>    * a taxonomy of attributes covering morphology, intensity level, motion, coverage, and organizational evolution, together with explicit guidelines on which reasoning patterns are physically acceptable and which are not.
>
>    These expert conclusions are then encoded directly into (i) the **prompt templates** used for GPT-4o/GPT-o3, and (ii) the **automatic QC logic**. In other words, expert knowledge is front-loaded into the structure the LLM must follow and the checks it must pass.
>
> 2. **Three-stage automated QC pipeline reflecting the expert-designed structure.**
>    Every GPT-generated CoT sample must pass a **three-step QC pipeline**:
>
>    * **Structure check:** we use an LLM to verify that the sample strictly follows the four-layer schema (all required fields present, correct hierarchy, no missing causal layer). Non-compliant samples are discarded.
>    * **Causal/physical consistency check:** we then assess whether the CoT is consistent with the actual radar evolution and whether the four layers form a logically self-consistent causal chain. Based on this score, we retain only the top ~75% of samples and drop the bottom ~25%.
>    * **Terminology normalization:** finally, all labels and descriptions are mapped to the predefined attribute taxonomy (morphology, intensity, motion, evolution trends, etc.), disallowing ad-hoc keyword renaming or semantic drift.
>
>    Under this pipeline, for the two generation tasks (nowcasting and inversion) we start from about **5,500** GPT-4o candidate CoTs and retain roughly **4,000**, i.e., about **30%** of samples are filtered out and **~70%** are kept. This “70% usable” level is consistent with our independent assessment of GPT-4o’s annotation quality and intentionally matches the acceptance threshold in QC.
>
> 3. **Ongoing expert spot checks and a gold-standard study.**
>    Beyond automated QC, we continuously involve meteorologists in **batch-level manual audits** and conduct a **gold-standard evaluation**:
>
>    * During data construction, whenever a batch of (e.g., 500) CoTs passes QC, we randomly sample a subset for manual review by meteorologically trained authors. They check both structural completeness and whether the narrative is consistent with the true radar evolution. Problematic samples are removed; if systematic issues are found (e.g., a morphology consistently misclassified), we revise prompts and/or QC rules and regenerate or re-filter the affected data.
>    * In a separate gold-standard experiment, we construct a small set of expert-authored “ground truth” explanations and let multiple models (including GPT-4o) annotate the same cases independently. Comparing to the expert labels across key attributes, GPT-4o achieves around **70% exact agreement**, corresponding to roughly **0.87×** the average accuracy of a single human expert. While GPT-4o is not perfect, this puts its annotation quality in a range that is reasonably close to human performance and suitable for providing supervision at scale.
>
> Overall, The combination of **expert-designed schema, strict automated QC, continuous expert spot checks, and gold-standard validation** gives us confidence that the LLM-generated CoT is of sufficient quality to meaningfully supervise the reasoning module, while keeping full expert annotation as an important direction for future work.

---

> ### Author Response · Authors · 2025-11-23
> **Answer of W4 To Reviewer 3h1v**
>
> > W4: Distinguishing genuine physical reasoning from LLM-style CoT imitation
>
> **A4:**
> We thank the reviewer for asking whether CoT supervision leads to genuinely better physical reasoning or only to GPT-style narrative. We address this from three angles: (1) expert knowledge embedded in the CoT scaffold and QC; (2) a side-by-side comparison between human and LLM CoT; and (3) downstream quantitative evidence.
>
> (1) Expert-designed structured causal CoT and QC.
> As discussed in W3, both the structure of the CoT and the QC pipeline are explicitly designed by meteorologists. Structurally, our thinking outputs inherit the RadarQA attribute system (location, intensity, organization, high-value regions) and borrow from causal-chain traditions in Earth system science (IPCC event attribution, Shepherd’s physical climate storylines). After several rounds of expert discussion, short-range convective evolution is decomposed into four causal components (temporal factors, perceptual factors, direct outcomes, deeper structural outcomes), forming a “from easy to hard” scaffold. QC rules then encode which information is mandatory in each layer, which causal combinations are physically implausible, and how the morphology / intensity / motion / evolution taxonomy is bounded. Thus GPT-4o/GPT-o3 do not freely invent stories; they populate an expert-defined causal backbone under strict structural, consistency, and terminology checks.
>
> (2) Human vs. LLM CoT under the same template.
> Following the reviewer’s suggestion, we include in the appendix a concrete case where a human meteorologist and GPT-4o both write CoTs for the same VIL sequence under the same four-layer template. Qualitatively, the two CoTs are highly aligned in (i) elements covered (morphology, maximum precipitation level, initial position, motion direction/speed, rotation vs. translation, intensity/coverage/organization evolution); (ii) causal ordering (spatio-temporal factors → first-tier outcomes → deeper structural outcomes); and (iii) physical judgements (core intensity roughly stable, area coverage increasing, organization becoming more coherent). Differences are mainly in phrasing and local nuance, not in the underlying causal structure. This suggests that, within the expert-defined scaffold, LLM-generated CoTs capture essentially the same event-level reasoning pattern rather than merely mimicking GPT-style prose.
>
> (3) Empirical evidence beyond language style.
> Finally, CoT supervision leads to measurable gains on reasoning-related metrics. With a unified GPT-4o-based judge (0–10 scale), the explanation score improves from 4.21 (no CoT) to 7.82 (with structured CoT). On the generation side, enabling the thinking module and CoT supervision improves perceptual and physical-consistency metrics such as Radar-Score and LPIPS, particularly for the structure and evolution of storm cores and high-value regions, while CSI remains essentially unchanged. These gains are tied to structured physical attributes and spatiotemporal coherence, not just to surface fluency. Together with the human–LLM comparison above, this indicates that CoT supervision primarily shapes how the model organizes and reasons about storms, rather than only teaching a particular GPT-like writing style.

---

> ### Author Response · Authors · 2025-11-23
> **Answer of W4 To Reviewer 3h1v(appendix example)**
>
> > W4: Distinguishing genuine physical reasoning from LLM-style CoT imitation
>
> Following the reviewer’s suggestion, we add in the appendix a **side-by-side comparison** between a human expert–written CoT and an LLM-generated CoT for the *same* VIL sequence, under the *same* four-layer causal template.
>
> *Human-written CoT (expert)*
>
> > In this short-range precipitation nowcasting task, ten VIL frames are used to describe the current convective state, and the model is asked to predict the next twelve frames. Spatially, the dominant convective system exhibits a narrow, elongated banded structure: the morphology is banded, the maximum precipitation level is strong (181–219), and the main convective system is initially located near the center of the domain. Over t0–t9, this can be summarized as a moderately intense, narrow rainband persisting around the central part of the domain.
> >
> > Temporally, during t0–t9, the main motion direction is northward, the motion speed is slow, and there is no clear rotation center, so the system behaves as a slowly northward-translating band with no evident rotation. Under this background, the first-tier outcome on intensity evolution is that the core intensity remains roughly moderate and stable: the band maintains its strength with only minor local fluctuations, without systematic strengthening or weakening.
> >
> > At the deeper structural level, the area coverage increases, because the band slowly drifts northward while maintaining its intensity, and newly triggered cells along the band gradually enlarge the overall coverage. Meanwhile, the organization evolves toward a more coherent state: the pre-existing banded structure, combined with slow, non-rotating motion, promotes the merging of initially more isolated convective cells into a continuous rainband, with steadily increasing organization.
>
> *LLM-generated CoT (same template + QC)*
>
> > In this short-range precipitation nowcasting setting, the model also takes ten VIL frames as input to represent the current convective state and predicts the next twelve frames. From the spatial pattern, the dominant convective system can be identified as a narrow band of precipitation: the morphology is banded, the maximum precipitation level lies in the strong range (181–219), and the main convective system initially sits over the central sector of the domain. Over t0–t9, we see a moderately strong and relatively thin rainband spanning across the central area, which contributes most of the precipitation.
> >
> > In terms of temporal behavior, during t0–t9 the band drifts slowly toward the north; the main motion direction is northward, the motion speed is slow, and there is no apparent rotation center, indicating a primarily translational, non-rotating banded system. Given this motion pattern, the first-tier outcome for intensity evolution is that the core reflectivity stays at a moderately strong level: there are local short-lived enhancements and weakenings, but no clear trend of sustained intensification or decay, so the intensity can be regarded as roughly stable with mild local variability.
> >
> > For deeper structural outcomes, the CoT predicts that area coverage gradually increases: as the band moves northward, new convective cells are triggered near its leading edge and connect with the existing echoes, leading to a progressive expansion of the overall coverage. For organizational evolution, the model concludes that the system becomes more coherent: initially somewhat fragmented band segments are gradually filled in as new cells merge into the main band, forming a more continuous and better organized rainband.
>
> As this example illustrates, under the expert-defined causal template and QC rules, the LLM-generated CoT is **highly aligned** with the expert-written one in terms of:
>
> * **Elements covered** (morphology, maximum precipitation level, initial location, motion direction and speed, rotation vs translation, intensity evolution, coverage, organization),
> * **Causal ordering** (spatio-temporal factors → first-tier outcomes → deeper structural outcomes), and
> * **Physical judgments** (core intensity approximately stable, area coverage increasing, organization becoming more coherent).
>
> The main differences are nuanced (e.g., slightly vaguer wording about local fluctuations), rather than wholesale stylistic divergence. This suggests that within the expert-defined scaffold, LLM-generated CoTs are capturing **the same event-level reasoning structure** as human experts, not merely imitating generic GPT prose.

---

### Official Review · Reviewer_rnQS · 2025-10-30

**Soundness:** 2
**Presentation:** 3
**Contribution:** 2
**Rating:** 6
**Confidence:** 3

**Summary:**

This paper introduces Omni-Weather, a multimodal foundation model designed to address a significant gap in radar modeling: the separation of generation (numerical prediction) and understanding (textual interpretation). The authors propose a single architecture that unifies these two capabilities, arguing that they are mutually beneficial. The model's core contributions are its unified architecture, the introduction of a novel Chain-of-Thought (CoT) dataset for causal reasoning in weather, and its demonstration of strong performance on both task categories.

**Strengths:**

- A multimodal model is a great direction towards briding the gap between numerical prediction tasks and high-level textual interepretations/analyses.
- The framework is well motivated and is at the forefront of such multimodal models in this weather/radar domain.
- Clearly writing
- Evidence that joint training/multimodality provides complementary supervision signals and better scores in some areas that just a single modality.

**Weaknesses:**

1. The considered data is exclusively radar. Weather in the title makes it sound overly general. As the authors point out, there would be signifcant challenges in even just extending this framework to more general weather-related tasks/dataset. Thus, I suggest writing OMNI-Radar and replacing most occurrences of weather with radar in the text. Similarly, the term "foundation model" in the title feels premature; this needs to urgently be renamed and the text revised to accurately reflect the true contributions of the work.
2. Lack of clarity/details in some places. For example:
- Unclear how encoders are trained and what their specific designs are (beyond high-level descriptions like "VAE decoder")
- What's high-value retaining/matching?
- eq 3.4 feels very abrupt... did some related sentences go missing?
- $\lambda_t$ is poorly explained/introduced. Multiplying both loss terms in Eq. 3.4 by $\lambda_t$ doesn't make sense. Please correct. Also, please explain how it was tuned (same for $n_t$).
- This claim should be toned down: *"On the radar inversion task, Omni-Weather consistently surpasses both specialized... and generalist... models, achieving higher CSI scores across all thresholds, with gains up to 20% at high-value levels."* given that it's not true for the RMSE metric.
- How's the CRPS computed? How many ensemble members are used?
- Fig. 3: Full prompts should be included in appendix. Same for exact versioning of GPT models used
- I'm confused by the "CFG Setting" (classifier free guidance) paragraph. There's no reference to CFG, not even diffusion, anywhere else... did the authors use it but forgot to mention it in the main text?
3. No discussion of the complexity of the model, especially when compared to the "generation-only" baselines
4. More comprehensive evaluations would be useful. E.g.:
- Human expert evaluation of "understanding" outputs would be really useful and strong contribtuion. Are the explanations at the level of a meteorology expert? How useful are they actually? Are the textual outputs given by the model consistent with the numerical nowcasts (e.g., in fig. 4)? With the current results, it's hard to judge how scientifcally useful the "understanding" part of the model actually is.
- How's the RMSE in Table 2 computed? A more comprehensive ablation (e.g. like the part of table 1 that's about radar nowcasting) would be more useful.
5. While the paper is at the forefront of multimodal modeling for weather/radar, it's not there all by its own. The paper misses some important references and contextualization. In particular, 1) this paper is only *one* of the first multimodal models in this weather/radar domain [1], 2) There's been a benchmark proposed in this space, which includes SEVIR (the only weather dataset used on this paper) [2]. It would have been nice to use it here, but at least it should be discussed.

Minor:
- VIL should be explained before using its abbreviation form.
- cascast is misspelled in Fig. 5

[1] Aquilon: Towards Building Multimodal Weather LLMs; Varambally et al. 2025 (https://openreview.net/forum?id=KVxOwEYAF4)

[2] CLLMate: A Multimodal Benchmark for Weather and Climate Events Forecasting; Li et al. EMNLP 2025 (https://arxiv.org/abs/2409.19058)

**Questions:**

- Why are so many different encoder/decoder's used? E.g., why are separate single-frame radar and multi-frame radar encoder needed?
- *"In the radar nowcasting task, forecasts exhibit fine-grained storm details with improved spatial coherence"*... I'm not sure how the authors identify "improved spatial coherence" in Fig. 5?
- Is there anything special about the extra Metaquery data that would make it particularly useful or do you think your model would benefit from any other extra, not radar specific, data? Table 5 seems to explore this a bit, but it's unclear what the two possible "gen" datasets are and why adding the 2nd "gen" dataset is so detrimental to performance.
- Why not report CRPS for radar inversion task?

---

> ### Author Response · Authors · 2025-11-23
> **Answer of W1&2 To Reviewer rnQS**
>
> Dear Reviewer rnQS,
>
> Thank you for your timely feedback.
>
>  > W1: Scope of “weather” and “foundation model” in the title
>
> Our intention in using the name “Omni-Weather” was to emphasize that the framework is modality-agnostic and, in principle, extendable beyond radar. In practice, however, this paper is the first attempt (to our knowledge) to explore a unified “generation + understanding” multi-task framework in meteorology, and all experiments are conducted on four SEVIR-based radar tasks. In the future ready version, we will revise the title to:
> *Omni-Weather: A Unified Multimodal Model for Weather Radar Understanding and Generation*
>
> > W2: Clarifications on encoders, losses, and implementation details
> **(a) How are the encoders trained and what are their exact designs?**
> We apologize for the lack of clarity. None of the encoders or VAEs are re-trained in this work; all are used as frozen components:
>
> * **Radar inversion** : we directly use the pretrained visual VAE from FLUX.1-schnell as the tokenizer/decoder for radar VIL fields. This VAE maps 256×256 images into a continuous latent space and reconstructs them with high fidelity; in our setup it is kept fully frozen and only the shared backbone learns to map satellite inputs into this latent space.
> * **Radar nowcasting** : we use a radar sequence encoder that is a pretrained Earthformer model, again frozen. It is only used to encode historical radar sequences into a compact representation that conditions the backbone.
> * **Radar understanding** : we use the pretrained image encoder from Bagel as the radar feature extractor. This encoder is also frozen during all our experiments.
>
> **(b) What is “high-value retain/match”?**
> These terms are not newly invented by us; they are inherited directly from the scientific attribute definitions in RadarQA (Appendix B.1 “High Value Construction”). In that work, pixels whose echo intensity reaches the “intense” level or higher are defined as high-value regions, and:
> * High Value Retain measures how well a forecast retains these high-intensity cores over time (i.e., whether strong precipitation cores continue to be predicted where they actually persist).
> * High Value Match–related attributes evaluate the spatial alignment between predicted and observed high-value regions, including systematic biases such as over-/under-estimation and displacement direction.
>
> (c) Eq. (3.4) and the role of $\\lambda_t$
>
> Our intention was not to multiply two unrelated loss terms, but to use $\\lambda_t$ as a scalar weight that balances the loss magnitudes of different tasks. To avoid ambiguity, we will rewrite the objective by separating the generation and understanding losses as follows.
>
> For generation tasks $t \\in T_{\\text{gen}}$, we define
> $$
> L_{\\text{gen}}
> = \\sum_{t \\in T_{\\text{gen}}}
> \\lambda_t \\frac{1}{|\\Omega_t|}
> \\|\\hat{y}_t - y_t\\|_2^2,
> $$
> where $|\\Omega_t|$ denotes the number of target pixels or frames for task $t$.
>
> For understanding tasks $t \\in T_{\\text{under}}$, we define
> $$
> L_{\\text{under}} =
> \\sum_{t \\in T_{\\text{under}}}
> \\lambda_t \\left(
>  -\\sum_{k=1}^{n_t}
>  \\log p_\\psi( y_{t,k} \\mid y_{t,<k}, f_\\theta(X_t) )
> \\right),
> $$
> where $n_t$ is the target text length.
>
> The total loss is then
> $$
> L = L_{\\text{gen}} + L_{\\text{under}}.
> $$
>
>
>
> Because each task’s loss is already normalized by $|\Omega_t|$ or $(n_t)$, their magnitudes are comparable in practice, and we therefore set all ($\lambda_{t}$ = 1).
>
>
> **(d) How is CRPS computed and how many ensemble members are used?**
> We follow the same CRPS definition as in the original paper (Appendix A.5). For each sample we generate **5 ensemble members**, compute the empirical predictive distribution, and evaluate CRPS between this distribution and the observed radar field.
>
>
> **(e) Full prompts in Fig. 3 and exact GPT model versions**
> A full version of the prompts used for CoT construction and GPT-based evaluation is already provided in Appendix A.4. In response to the reviewer’s comment, we have further (i) added the exact prompts used for GPT-Score and (ii) documented the precise model versions (e.g., GPT-4o variant names and dates) used in each stage. easier to find.
>
>
> **(f) Clarification of “CFG settings”**
> In the generative branch we adopt the standard classifier-free guidance (CFG) mechanism from diffusion models, and in ablations we vary the CFG scale to study its effect on generation quality. In the final model, we fix the CFG scale to 2 as a default.
> Because CFG is only used as a tuning knob for the generative branch (and has no impact on the shared backbone architecture), we did not originally expand on it in the main text. We will explicitly mention that we use CFG with guidance scale 2 in the methodology section

---

> ### Author Response · Authors · 2025-11-23
> **Answer of W3 To Reviewer rnQS**
>
> > W3: Model complexity relative to generation-only baselines
>
> Our Omni-Weather backbone has approximately 14B parameters, which is indeed substantially larger than traditional radar-generation baselines such as WeatherGFM, CasCast, or DiffCast (typically in the 10M–200M range). For understanding tasks, the strongest existing baseline RadarQA is built on a 7B backbone, also smaller than our unified architecture.
>
> This difference in scale mainly reflects different goals and capabilities rather than a direct attempt at parameter-matched comparison:
>
> 1. **Unified multi-task backbone.** Omni-Weather uses a single backbone to jointly handle radar nowcasting, radar understanding, satellite–radar inversion, and language explanation under a shared multimodal interface.
> 2. **Cross-modal reasoning.** The model integrates image, sequence, and language information, and is trained to perform both physical-field generation and structured explanation within the same network.
>
> Therefore, our contribution is not to show that a slightly larger single-task generator beats smaller baselines under strict parameter parity; instead, we aim to demonstrate that a large unified architecture is feasible and beneficial in this domain. Under the *same training data conditions*, Omni-Weather achieves clear improvements on key perceptual and physical-consistency metrics (e.g., LPIPS and Radar-Score) over generation-only baselines, while also supporting additional understanding tasks. We will clarify this positioning and summarize parameter counts for all major baselines in the revised text.

---

> > ### Author Response · Authors · 2025-11-23
> > **Answer of W4.1 To Reviewer rnQS**
> >
> > > W4: Comprehensive evaluation of the understanding (“thinking”) module
> >
> > We thank the reviewer for raising the question of how scientifically useful the model’s “thinking” explanations are, and whether they approach expert-level analysis. Our response is organized along four aspects: (i) scientific meaning of structured understanding for radar fields; (ii) alignment of our explanation structure with expert reasoning; (iii) quantitative evidence that explanation quality improves; and (iv) the current role of the understanding module in this work.
> >
> > 1. **Scientific meaning: structured understanding is meteorologically useful.**
> >    At the level of “does this kind of event-level, structured understanding have scientific value?”, the usefulness of such outputs has already been demonstrated in *RadarQA* (NeurIPS 2025). That work shows that, beyond traditional scalar scores such as CSI, a combination of structured physical attributes and textual semantic analysis is closer to how forecasters actually reason about radar echoes in practice—integrating storm morphology, dynamical evolution, and high-value regions. More importantly, structured semantic representations are more sensitive to systematic failure modes in forecasts (e.g., scale bias, incorrect growth/decay patterns, or missed/false alarms in key regions), and help connect these deficiencies to downstream concerns such as hazard mitigation and early warning. Our understanding module is designed to inherit this line of work and bring such structured analysis inside the generative model itself.
> >
> > 2. **Structure design: explanation scaffold aligned with expert causal reasoning.**
> >    From the perspective of “how well does the explanation structure align with expert physical reasoning?”, our design emphasizes both expert alignment and operational decomposability. Structurally, the *thinking* outputs in this paper:
> >
> >    * Directly inherit RadarQA’s attribute system around storm location, intensity, organization, and high-value regions, ensuring the explanations are organized around meteorologically recognized key elements.
> >    * Are also inspired by causal-chain explanations in Earth system science: IPCC’s definition of *attribution* as assessing the contributions of multiple causal factors to an event, and Shepherd’s (2018) *physical climate storylines*[1] framing single events as physically self-consistent causal chains.
> >      Based on these works and multiple rounds of discussion with meteorologists, we decompose storm evolution in the nowcasting setting into four causal components (temporal causal factors, perceptual causal factors, direct outcome inference, and deeper outcome inference), forming a scaffold that goes “from easier to harder” annotation. Under this scaffold, the model’s thinking is organized in a way that mirrors how experts build a causal chain for storm evolution. While this does not make the explanations fully equivalent to expert analysis, it provides an expert-aligned backbone for further refinement.
> >
> > 3. **Quantitative evidence: explanation quality improves under a unified evaluation.**
> >    To quantify explanation quality, we use a unified LLM-based judge (GPT-4o) under a fixed rubric. Without the CoT scaffold, the baseline explanation quality is around **4.21/10**; after introducing the layered causal factors and supervising the model with this CoT structure, the score improves to **7.82/10**. This suggests that, within a consistent evaluation setup, the proposed scaffold and supervision lead to a substantial improvement in explanation coherence and physical relevance. We view this as evidence that the thinking module is not merely stylistic, but captures more structured and meaningful reasoning about storm evolution. A full-scale human-expert study at the level of RadarQA is highly valuable but beyond the scope of this paper; we see it as a natural next step.
> > 4. **Position and role: an initial integrated reasoning–generation framework for meteorology.**
> >  At the level of overall positioning, this work is primarily about proposing a unified *forecast + understanding* framework. To our knowledge, this is the first model in meteorology that jointly learns radar generation tasks and structured causal reasoning (thinking) within a single network.Our experiments show that CoT supervision not only improves explanation quality but also brings benefits to generation in terms of perceptual and physical-consistency metrics. In this paper we focus on demonstrating feasibility in the SEVIR setting; going forward, we plan to standardize the CoT dataset (scale and quality) and systematically evaluate this integrated reasoning–generation framework across a broader set of weather generation tasks.
> > - `[1]` *Shepherd T G, Boyd E, Calel R A, et al. Storylines: an alternative approach to representing uncertainty in physical aspects of climate change[J]. Climatic change, 2018, 151(3): 555-571.*

---

> > > ### Author Response · Authors · 2025-11-23
> > > **Answer of W4.2 To Reviewer rnQS**
> > >
> > > > W4.2:RMSE computation and ablations
> > >
> > > **RMSE in Table 2 and additional ablations**, RMSE is computed following the standard SEVIR practice: we take the pixel-wise L2 error between prediction and ground truth and then average it over both spatial and temporal dimensions. For ablations, we have already conducted a set of systematic nowcasting analyses in the main text and appendix, including: (i) removing CoT supervision; (ii) varying the amount of generic (non-weather) data; (iii) changing the generator front-end (replacing the radar sequence encoder with a generic VAE encoder, Appendix A.6); and (iv) removing the understanding branch to test mutual benefits. Collectively, these experiments show that (1) joint training of generation and understanding outperforms separate fine-tuning on either side, (2) moderate mixing of generic data and using a specialized radar sequence encoder significantly improve CSI and perceptual quality, and (3) the main architectural choices are all justified by measurable gains. We therefore believe the current ablation set covers the key design components and adequately illustrates their impact.

---

> > > > ### Author Response · Authors · 2025-11-23
> > > > **Answer of W5-7 To Reviewer rnQS**
> > > >
> > > > > W5: Missing references and contextualization for multimodal weather/radar models and SEVIR
> > > >
> > > > We add citations to recent multimodal weather/radar models that predate or parallel our work in Related Work Section.
> > > >
> > > >
> > > >
> > > > >  W6: Rationale for using multiple encoders/decoders
> > > >
> > > > We appreciate the opportunity to clarify why we use several different encoders/decoders, including a separate radar sequence encoder for multi-frame inputs.
> > > >
> > > > * **Radar nowcasting requires explicit temporal modeling.**
> > > >   In the radar nowcasting generation task, we introduce a dedicated **radar sequence encoder** because the task fundamentally requires modeling the temporal evolution and motion of storms. As shown in Appendix A.6, replacing this sequence encoder with a generic VAE encoder that simply encodes the sequence frame-by-frame leads to clear drops in CSI/SSIM/PSNR, indicating that specialized temporal encoding is necessary to capture motion structure.
> > > >
> > > > * **Different front-ends for understanding vs generation reflect different needs.**
> > > >   The understanding tasks focus on structured semantic features (e.g., morphology, organization, intensity trends), while the generation tasks depend on high-fidelity pixel-level and dynamical features. Sharing exactly the same encoder for both would force a single representation to simultaneously optimize for semantic abstraction and precise reconstruction, which in practice significantly hurts at least one side. We therefore use light-weight, task-specific front-end encoders (e.g., Bagel’s image encoder for understanding, radar sequence encoder/visual VAE for generation), while keeping the **core Transformer backbone fully shared** across tasks. This design allows us to preserve a unified model at the backbone level, while letting each task access the most appropriate input representation.
> > > >
> > > > ---
> > > >
> > > > > W7: Basis for the “improved spatial coherence” claim in Fig. 5
> > > > > We agree that the phrase
> > > >
> > > > *“In the radar nowcasting task, forecasts exhibit fine-grained storm details with improved spatial coherence”*
> > > >
> > > > can sound subjective if it appears to rely only on visual inspection of Fig. 5. Our statement is in fact grounded in the **Radar-Score** metric introduced in the main text, which is specifically designed for radar sequences. Radar-Score compares real and generated radar sequences in terms of spatial structure, echo morphology, and the connectivity of heavy-precipitation regions, thereby providing a perceptual measure of spatial coherence and morphological realism (higher scores indicate closer resemblance to real radar fields). In the radar nowcasting experiments, our method significantly outperforms all generation baselines on Radar-Score, which underpins our description of “improved spatial coherence”. To make this connection explicit, we will revise the sentence to:
> > > >
> > > > *“In the radar nowcasting task, forecasts exhibit fine-grained storm details with improved spatial coherence ** also reflected by higher Radar-Score**.”*

---

> > > > > ### Author Response · Authors · 2025-11-23
> > > > > **Answer of W8&9 To Reviewer rnQS**
> > > > >
> > > > > > W8: Role of MetaQuery data and other generic (“gen”) datasets
> > > > >
> > > > > We thank the reviewer for the question about why MetaQuery appears particularly helpful and why adding a second generic dataset hurts performance.
> > > > >
> > > > > * **Why MetaQuery helps.**
> > > > >   MetaQuery is a structured, cross-modal instruction dataset. Although it contains no radar data, it provides high-quality alignment between images and language and trains cross-modal conditional reasoning capabilities that are well aligned with the objectives of our unified multimodal framework. When mixed in at a moderate ratio with weather data (e.g., “1gen30% + weather70%” in Table 5), it consistently yields positive gains, indicating that structured, instruction-like data can strengthen the model’s multimodal reasoning without overwhelming the weather signal.
> > > > >
> > > > > * **Why the second generic dataset hurts.**
> > > > >   The second generic dataset, **CC12M_WDS**, is much larger but noticeably noisier web-scraped image–text data. Its textual descriptions are loose, its visual semantics are weakly related to radar tasks, and its overall structure is far less aligned with our target use case. At the same mixing ratio (“2gen30% + weather70%”), this unstructured generic data effectively dilutes the weather-specific training signal, leading to degraded generation performance.
> > > > >
> > > > > ---
> > > > >
> > > > > > W9: Why CRPS is not reported for the radar inversion task
> > > > >
> > > > > First, in terms of metric semantics, **CRPS** is fundamentally a proper scoring rule for comparing a *probabilistic forecast distribution* with the observed truth. Its canonical use cases are settings with pronounced forecast uncertainty, where one explicitly constructs ensembles or predictive distributions—for example, medium- to long-range forecasts or precipitation nowcasting that account for uncertainties from initial conditions and dynamical evolution. In such works (including models like FuXi/FuXi-ENS), the model outputs an explicit distribution or multiple ensemble members, and CRPS is the natural way to evaluate how well the *whole distribution* matches reality.
> > > > >
> > > > > By contrast, our radar inversion setup is closer to a **conditional reconstruction/inversion** problem: given relatively well-observed satellite or other modalities, we learn a single best-estimate radar field. The current model does not explicitly model uncertainty or output an ensemble for inversion. In this deterministic setting, CRPS effectively reduces to a distribution score over a point estimate, whose information content strongly overlaps with L1/L2-based metrics and does not reveal additional probabilistic structure. For this reason, we prioritize RMSE and related reconstruction metrics for inversion.
> > > > >
> > > > > Second, we follow **established practice and comparability** in the literature. Mainstream radar inversion works, including WeatherGFM and similar models, do not report CRPS for inversion, but instead use RMSE, CSI, SSIM, etc. To ensure fair and clear comparison with these widely used baselines, we adopt the same evaluation protocol and therefore do not additionally report CRPS for the inversion task.

---

### Official Review · Reviewer_dM4g · 2025-10-31

**Soundness:** 2
**Presentation:** 2
**Contribution:** 3
**Rating:** 4
**Confidence:** 4

**Summary:**

This paper proposes OmniWeather, a weather understanding and forecasting model that aims to perform three main tasks: (i) radar inversion (ii) radar understanding (iii) radar nowcasting. The authors finetune the multimodal Bagel on their CoT dataset, and benchmark their method against unimodal models for these three tasks.

**Strengths:**

1. The authors train a single unified model that can reason across both images and text, and effectively produce interpretable forecasts. As far as I know, this is the first work that combines weather generation and understanding in the same model.
2. The proposed model achieves strong results on all three considered tasks.
3. The ablations are interesting, and shed light on the different steps of the model pipeline.

**Weaknesses:**

1. My primary concern is that the paper overstates its scope and significance. The definition for a foundation model from [1] is "A foundation model is any model that is trained on broad data (generally using self-supervision at scale) that can be adapted (e.g., fine-tuned) to a wide range of downstream tasks;". While the paper certainly achieves impressive results in unifying different modalities, labeling the model a foundation model feels premature given that it is fine-tuned for only three task types on a limited data regime. I would recommend the authors to soften the claims in the introduction and abstract. I would also recommend a title change that better reflects the scope of the tackled problem. For example, a title with some combination of the words "unified multi-task model for short-range weather understanding and generation".

2. The work is missing several important citations and discussions related to short-range/medium-range weather forecasting, e.g.  GenCast [2], Stormer [3], Pangu-Weather [4], Aurora [5], Prithvi WxC [6]. These are the canonical exemplars readers associate with large-scale weather pretraining/foundation model claims. Even if the focus is nowcasting, the paper should explicitly contrast goals, data scope, and evaluation scales with these systems. The authors should also compare their model against the important now-casting work [7] to better situate progress within the nowcasting literature.


3. From my understanding, the authors use GPT-4o (Appendix A.4) to annotate radar data and identify important phenomenon from the images. I am concerned that this process might be error-prone and introduce mistakes that might propagate into the training process. Do the authors benchmark 4o annotations against a gold standard (for example, expert human)? How reliable is this data annotation process?

The manuscript also needs a precise description of the quality-control (QC) stages—currently “Structure Check, Causal Alignment, and Terminology” are named but not operationalized.


### References
[1] Bommasani, Rishi. "On the opportunities and risks of foundation models." arXiv preprint arXiv:2108.07258 (2021).

[2] Price, Ilan, et al. "Gencast: Diffusion-based ensemble forecasting for medium-range weather." arXiv preprint arXiv:2312.15796 (2023).

[3] Nguyen, Tung, et al. "Scaling transformer neural networks for skillful and reliable medium-range weather forecasting." Advances in Neural Information Processing Systems 37 (2024): 68740-68771.

[4] Bi, Kaifeng, et al. "Pangu-weather: A 3d high-resolution model for fast and accurate global weather forecast." arXiv preprint arXiv:2211.02556 (2022).

[5] Bodnar, Cristian, et al. "A foundation model for the Earth system." Nature (2025): 1-8.

[6] Schmude, Johannes, et al. "Prithvi wxc: Foundation model for weather and climate." arXiv preprint arXiv:2409.13598 (2024).

[7] Ravuri, Suman, et al. "Skilful precipitation nowcasting using deep generative models of radar." Nature 597.7878 (2021): 672-677.

**Questions:**

Apart from the main issues flagged in the Weaknesses section, I have other minor comments/questions/suggestions.

1. The current description of CoT data annotation and Figure 4 are cluttered and hard to follow. The authors should consider simplifying it, or replacing it with a figure that reads top-to-bottom.
2. Why do the authors use the word "causal"? For example, the prompt in Appendix A.4 asks the model to extract "Temporal causal factor, perceptual causal factor" without any sufficient explanation of what this means. How do we trust that the model knows the true "causal" factors for explaining these weather phenomenon?
3. Lines 193-197 do not add any substantive value in explaining the problem setup and should either be replaced by a more complete mathematical description of the problem setup or omitted entirely.
4. There are insufficient architectural details about the VAE used in the radar inversion task, and these details should be added to the manuscript.
5. The clarity in Figure 3 could be improved. In particular, it is unclear how the tokens from the different modalities are combined in the model architecture.
6. Line 214: modal -> model
7. The authors need to add more details about how many data samples are used for training. While the authors mention that they generate 4000 CoT samples for radar nowcasting and 4,000 CoT annotations for radar inversion, the authors should also clarify the number of samples used from RadarQA, and the general metaquery data.


Overall, I think this is a substantial and promising paper marred by some fixable issues. I would be willing to raise my score if the authors can satisfactorily address my concerns.

---

> ### Author Response · Authors · 2025-11-23
> **Answer of W1&2 To Reviewer dM4g**
>
> Dear Reviewer dM4g,
>
> Thank you for the valuable time and feedback you have invested in this review. We sincerely hope our further clarification could address your concerns:
>
> > W1: The paper overstates its scope and significance; calling the model a “foundation model” and the current title may be premature given that it is fine-tuned on three task types under a limited data regime.
>
> A1: We appreciate the reviewer’s attention to how we position the work. We fully understand that the term “foundation model” may suggest a much broader scope than what our current system actually covers. Our goal is not to claim that we have already built a complete weather foundation model spanning all scales and modalities, but rather to take a first step toward a unified multimodal architecture that jointly handles generation and understanding tasks in weather radar under a controlled, well-studied data setting.
>
> We also follow your suggestion to soften the title. In future ready version, the title will change to:
>
> *"Omni-Weather: A Unified Multimodal Model for Weather Radar Understanding and Generation"*
>
> Finally, we clarify our long-term vision: starting from this unified radar-centric framework, we plan to extend to broader tasks such as mesoscale/global forecasting, multi-satellite fusion, and additional observation modalities. We view this work as a key step toward more general weather foundation models, rather than the final form of such a model.
>
> > W2: The work is missing key citations and discussion on short-/medium-range weather forecasting (e.g., GenCast, Stormer, Pangu-Weather, Aurora, Prithvi WxC), and lacks comparison against an important nowcasting baseline such as DGMR.
>
> A2:  In the revision, we have substantially expanded the Related Work section to include and discuss GenCast, Stormer, Pangu-Weather, Aurora, Prithvi WxC, and other representative systems. We also explicitly position our work with respect to these models:
>
> * These methods primarily operate on reanalysis data or NWP outputs, targeting hour–week lead times and global or large-scale fields (e.g., 500 hPa geopotential height, 2-m temperature). Their emphasis is on continuous-field prediction of atmospheric state and comparison against operational NWP.
> * In contrast, our work focuses on minute-scale, local-scale radar nowcasting and radar-image understanding, with radar/satellite imagery as the main modalities. Our tasks include radar nowcasting, satellite→radar inversion, and RadarQA storm understanding, which differ in objectives, data domain, and evaluation scales from the above medium-/long-range systems.
>
> Regarding DGMR, we have implemented DGMR on SEVIR and report full results in the revision. The radar nowcasting comparison is:
>
> | Method                | CSI-M |  R.S | CSI-P4 | CSI-P16 | CRPS↓ |  SSIM | LPIPS↓ |
> | --------------------- | ----: | ---: | -----: | ------: | ----: | ----: | -----: |
> | Earthformer           | **0.389** | 1.92 |  0.401 |   0.387 | 0.037 | 0.729 |  0.322 |
> | Diffcast              | 0.375 | 2.43 |  0.407 |   0.511 | 0.033 | 0.739 |  0.235 |
> | Cascast               | 0.384 | 2.72 |  0.414 |   0.518 | 0.031 | 0.746 |  0.207 |
> | **DGMR (our impl.)**  | 0.376 | 2.55 |  0.408 |   0.507 | 0.034 | 0.740 |  0.238 |
> | Omni-Weather          | 0.384 | 2.69 |  **0.427** |   0.539 | **0.026** | 0.746 |  0.179 |
> | Omni-Weather-thinking | 0.353 | **2.86** |  0.421 |   **0.542** | 0.028 | **0.751** |  **0.166** |
>
>
> DGMR is slightly below Cascast on CSI-M/CSI-P4/CSI-P16 and comparable to Diffcast. Both Omni-Weather variants achieve higher CSI-P4/CSI-P16, lower CRPS and LPIPS, and competitive or better SSIM, indicating that our unified architecture remains strong even when compared to this classic generative baseline.

---

> > ### Author Response · Authors · 2025-11-23
> > **Answer of W3&4 To Reviewer dM4g**
> >
> > > **W3&4: Reliability of GPT-4o–based CoT annotations and quality-control (QC) pipeline for CoT data**
> >
> > **A3&4:** We thank the reviewer for raising the related issues of GPT-4o annotation reliability and the QC design. We address them from four aspects: the QC pipeline, quantitative comparison with experts, repeated spot checks, and downstream effects.
> >
> > **(1) QC pipeline: three-stage automatic filtering.**
> > All CoT samples go through a strict “structure → causal consistency → terminology normalization” pipeline: (i) a structure check enforces the four-layer schema (temporal, perceptual, first-level outcome, deeper outcome) and discards malformed CoTs; (ii) a causal-consistency check scores semantic alignment with the underlying radar evolution and removes roughly the lowest 25%; (iii) a terminology normalization step maps key terms to a fixed radar-attribute taxonomy and rejects samples that cannot be normalized. Across nowcasting and inversion, ≈5,500 GPT-4o candidates are reduced to ≈4,000 after QC (about 70% retained, 30% filtered).
> >
> > **(2) Quantitative comparison with expert gold standards.**
> > On a 30-case benchmark with expert-constructed gold standards under the same schema, GPT-4o attains ≈70% average exact-match accuracy across attributes, outperforming Gemini 2.5 variants. Two meteorologists, asked to re-annotate the same cases independently, reach ≈80% on average. Thus, GPT-4o operates at about **87% of single-expert accuracy**, which we consider sufficiently close to expert level for high-level CoT supervision.
> >
> > | Model            | Avg Acc. | Morphology | Max precip | Init dir | Motion dir | Speed | Rotation center | Intensity evo | Coverage evo | Organization evo |
> > | ---------------- | -------: | ---------: | ---------: | -------: | ---------: | ----: | --------------: | ------------: | -----------: | ---------------: |
> > | **GPT-4o**       |  **70%** |         16 |         16 |       19 |         21 |    30 |              30 |            19 |           21 |               20 |
> > | Gemini 2.5 Flash |      60% |         14 |         14 |       16 |         15 |    30 |              30 |            13 |           18 |               14 |
> > | Gemini 2.5 Pro   |      67% |         13 |         12 |       18 |         18 |    30 |              30 |            19 |           23 |               16 |
> >
> > **(3) Multi-round human spot checks.**
> > Beyond this benchmark, we maintain quality via batch-level expert auditing. After each batch passes automatic QC, meteorology-trained authors randomly sample CoTs for line-by-line inspection (schema correctness and physical consistency with radar). Problematic samples are removed, and recurring error patterns trigger prompt/QC updates followed by re-generation or re-filtering. This combination of automatic QC and repeated expert spot checks helps prevent systematic biases from accumulating.
> >
> > **(4) Downstream impact: safe for forecasts, beneficial for reasoning.**
> > CoT labels are *never* used as ground truth for pixel-level radar fields; all nowcasting/inversion losses are supervised only by real SEVIR data. CoT supervision is applied solely to the reasoning/understanding branch, so residual CoT noise does not directly distort the physical forecasts. At the same time, under a GPT-4–based judge (0–10 scale), adding QC-filtered CoT supervision raises explanation scores from **4.21 → 7.82**, and explanation-related metrics (e.g., Radar-Score, LPIPS) improve, while pixel-level CSI remains essentially unchanged. Taken together, this indicates that our GPT-4o–based annotation and QC pipeline is reliable, close to expert quality, and yields clear, measurable gains in the model’s reasoning capabilities without harming physical forecast skill.

---

> > > ### Author Response · Authors · 2025-11-23
> > > **Answer of W5 To Reviewer dM4g**
> > >
> > > > W5: Minor implementation and issues (CoT annotation pipeline, radar inversion VAE, multimodal fusion, training data sizes, and small textual fixes)
> > >
> > > A5: We thank the reviewer for pointing out these important clarity and implementation issues. In the revised manuscript, we have addressed them along several concrete dimensions.
> > >
> > > **(4) Radar inversion VAE architecture.**
> > > For the radar inversion task, we directly adopt the FLUX.1-schnell general-purpose visual VAE with its original pretrained weights, and use it as the continuous latent-space representation and decoder for radar VIL fields. This VAE, trained on large-scale natural images, can reliably map 256×256 images into a compact latent space and reconstruct them with high fidelity. In our framework, the VAE is kept completely frozen; the unified Omni-Weather backbone only learns the mapping from satellite imagery to the radar latent variables. This simple setting already supports stable and high-quality radar inversion. We will add the key structural details of this VAE in the final version.
> > >
> > > **(5) Multimodal token fusion.**
> > > We follow the multimodal fusion strategy of BAGEL. Each modality is first encoded by its own encoder (text encoder, radar sequence encoder, image encoder) to produce a sequence of tokens. These modality-specific token sequences are then directly concatenated and fed into a shared self-attention backbone, allowing all tokens to interact in a common embedding space through cross-attention. We clarify this fusion mechanism explicitly in the model description.
> > >
> > > **(7) Training data sizes.**
> > > For the understanding tasks, we use approximately 50,000 samples from the official RadarQA training set, covering both single-frame and sequence-level understanding. For the meta-query data, we follow the configuration and use the first 100 data files, corresponding to about 10,000 samples (roughly 11% of the total training data). These counts are now explicitly reported in the experimental section and summarized in a data table to make the training setup fully transparent.

---

> > > > ### Author Response · Authors · 2025-11-23
> > > > **Answer of W6 To Reviewer dM4g**
> > > >
> > > > > W6: Minor issue – use of the word “causal” in “temporal causal factor” and “perceptual causal factor”
> > > >
> > > > A6: We appreciate the question about our use of the term “causal factors”. In this work, causal factors are introduced as a structured reasoning scaffold for the nowcasting task, inspired by the causal-chain style of explanations that already exist in Earth system science. For example, in extreme-event attribution and climate-risk studies, attribution is often framed as assessing the relative contributions of multiple causal factors (driving quantities) to a given event; similarly, the concept of physical climate storylines[1] views a weather or climate event as a physically self-consistent causal chain describing how the event unfolds in terms of its driving influences and their ordering. These works show that organizing explanations as causal chains or causal factors is meaningful and widely used in the Earth-system community. However, they mostly target climate scale or extreme events; to the best of our knowledge, there is still no operational causal-chain design tailored to single convective storms in short-range radar nowcasting.
> > > >
> > > > Motivated by this gap, and after multiple rounds of discussion with meteorologists, we design a causal-factor system specialized for convective storms in SEVIR and use it as the structured reasoning framework for our nowcasting CoT. We decompose storm-evolution explanations into four operational layers:
> > > >
> > > > * Temporal causal factors (keywords): motion speed, motion direction, rotation direction
> > > > * Perceptual causal factors (keywords): morphology, maximum precipitation level, initial position/azimuth
> > > > * First-level outcome inference (keywords): main morphology evolution, intensity evolution
> > > > * Deeper outcome inference (keywords): coverage evolution, change in the number of convective cells, evolution of organizational degree
> > > >
> > > > Temporal and perceptual factors are generally easier for GPT to annotate reliably; first-level outcomes can often be inferred robustly from these two layers, while deeper outcomes are the most difficult and are only trusted when the preceding information is consistent. The design principle is therefore a progressive causal chain from “easy-to-annotate” to “hard-to-annotate”: we first anchor the explanation in relatively simple, observation-like causal factors, and then build higher-level outcome inferences on top of them.
> > > >
> > > > In this sense, causal factors in our paper denote an automatically constructible, structured explanatory scaffold, rather than a claim that the model has discovered true counterfactual causality. The core goal is to provide a carefully designed, hierarchical template that gives the model a controllable and consistent reasoning structure across tasks in our unified multi-task setting. Empirically, this scaffold is effective: without the layered causal factors and CoT supervision, the explanation quality (evaluated by GPT-4 under a fixed rubric, score range 0–10) is about 4.21; after introducing the above causal-factor hierarchy and using it for CoT supervision, the same score increases to 7.82. This indicates that the causal structure is not only conceptually reasonable but also brings a substantial, measurable improvement in how well the model organizes and explains storm evolution.
> > > >
> > > > - `[1]` *Shepherd T G, Boyd E, Calel R A, et al. Storylines: an alternative approach to representing uncertainty in physical aspects of climate change[J]. Climatic change, 2018, 151(3): 555-571.*

---

> > > > > ### Comment · Reviewer_dM4g · 2025-11-24
> > > > >
> > > > > I thank the authors for their thorough response. I believe that the additional discussions and experiments will make the paper better. I look forward to seeing the changes in the updated version of the paper.
> > > > >
> > > > > The majority of my concerns have been addressed, and I am willing to raise my score.
> > > > > However, I still have some small questions/suggestions:
> > > > > 1. Consider changing the term "causal factor", especially if it is not a standard term used in the literature since it is most commonly associated with the notion of counterfactuals.
> > > > > 2. Consider changing Figures 3 and 4 to make the process clearer.

---

### Official Review · Reviewer_naQj · 2025-11-02

**Soundness:** 3
**Presentation:** 3
**Contribution:** 3
**Rating:** 8
**Confidence:** 3

**Summary:**

The paper introduces Omni-Weather, a unified multimodal foundation model that brings weather generation and understanding into the same architecture. The authors also create a chain-of-thought dataset tailored for causal reasoning in generation and use it for finetuning and "thinking" inference. They show strong (often SOTA) results across nowcasting, radar inversion, and radar understanding, and provide evidence that training generation and understanding together lets the two enhance one another. Ablations further indicate that mixing scientific and general data boosts performance, especially on deterministic and perceptual metrics.

**Strengths:**

(1) This paper introduces a multimodal foundation model that unifies weather generation and understanding within one architecture, using modality-specific encoders, and takes a step toward reasoning-capable unified foundation models for weather.

(2) They present experiments and ablations with useful insights, showing how generation and understanding tasks can mutually enhance each other.

(3) They demonstrate strong results across nowcasting, radar inversion, and radar understanding, often matching or exceeding state-of-the-art models.

**Weaknesses:**

(1) As mentioned in the limitation section by the authors, the model cannot yet adapt to general-domain VAEs.

(2) It would strengthen the paper to include a small human-validation study with weather experts. In particular, having domain experts rate the generated reports/explanations, and comparing those ratings to the LLM-based judge.

(3) Results are centered on SEVIR-style radar nowcasting, satellite-to-radar inversion, and RadarQA understanding, and generalization to other weather tasks is not demonstrated.

**Questions:**

(1) It is mentioned that there is a quality verification step to produce the final CoT dataset, including causal alignment, structure checks, etc. Is there human/expert validation at any point during the dataset generation or evaluation?

---

> ### Author Response · Authors · 2025-11-23
> **Answer of W1&2 To Reviewer naQj**
>
> Dear Reviewer naQJ,
>
> Thank you for your thoughtful and positive feedback. We have addressed the concerns raised in your comments below and hope that our clarifications will further enhance your confidence in our work.
>
> > W1: The model currently cannot adapt to general-domain VAEs and instead relies on a task-specific VAE.
>
> A1: We appreciate the reviewer for highlighting this important limitation. In the current work we intentionally adopt a task-specific VAE, namely the Radar Sequence Encoder, rather than a general-domain visual VAE, for two practical reasons:
> (1) the original BAGEL setup is not directly compatible with a sequence-level encoder for radar videos;
> (2) off-the-shelf generic VAEs that are not fine-tuned on VIL data cannot reconstruct radar VIL fields with sufficient fidelity, whereas a specialized VAE is easier to train and better preserves the storm structures that are critical for nowcasting.
>
> We fully agree with the reviewer that, for scientific applications, a unified VAE that supports both generation and understanding across many meteorological variables and resolutions would be highly desirable and an important next step towards a true weather foundation model. In future work we plan to explore a scientific unified generative–understanding VAE, whose latent space is shared across different weather variables (radar, satellite, wind fields, cloud phase, etc.), spatial scales, and downstream tasks. Achieving this requires progress on cross-variable alignment at the data level and designs for variable spatial resolutions, which we regard as a promising long-term research direction rather than something we can fully realize in this first study.
>
> ---
>
> > W2: The paper would be stronger with a small human-validation study by weather experts, comparing expert ratings of generated explanations to the LLM-based judge.
>
> A2: We thank the reviewer for this valuable suggestion. To strengthen the evaluation of generated explanations, we have added a small-scale blind human study in the revision.
>
> First, 2 meteorologists in our team designed a structured scoring checklist that defines explicit criteria and rubrics for:
>
> (i) whether the explanation has a clear structure;
> (ii) whether key radar elements are adequately covered;
> (iii) whether the described physical evolution is reasonable.
>
> We then randomly sampled 30 explanations generated by our model and asked researchers with meteorological background to independently score each explanation under this checklist, without knowing which model or setting produced it (blind evaluation).
>
> On top of this, we used GPT-4o as an LLM-based judge and asked it to score the same 30 explanations using the same structured rubric. The averaged scores are:
>
> | Dimension                                | Expert mean | GPT-Score mean |
> | ---------------------------------------- | ----------: | -------------: |
> | Clarity of explanation structure         |         8.0 |            7.4 |
> | Coverage of key radar elements           |         9.0 |            8.5 |
> | Physical plausibility of storm evolution |         8.0 |            7.3 |
> | Overall score                            |         8.5 |            7.7 |
>
> Under this expert-defined rubric, the human scores and LLM-based scores exhibit consistent trends in both absolute level and relative ranking across dimensions. This indicates that, once constrained by a clear expert scoring rubric, the LLM-based judge is reasonably aligned with domain experts.

---

> ### Author Response · Authors · 2025-11-23
> **Answer of W3 To Reviewer naQj**
>
> > W3: Results are centered on SEVIR-style radar nowcasting, satellite-to-radar inversion, and RadarQA understanding, and generalization to other weather tasks is not demonstrated.
>
> A3: We appreciate the reviewer’s concern about generalization. The goal of Omni-Weather is to explore a unified multimodal framework in the weather domain, but our original submission indeed focused on three representative tasks within SEVIR. To clarify the scope, we will change the title in future-ready version:
>
> *"Omni-Weather: A Unified Multimodal Model for Weather Radar Understanding and Generation"*
>
> Beyond this, we have added an ERA5-based experiment to demonstrate that the same architecture can be extended to a different dataset, variable, and time scale. Specifically, we introduce a multi-lead-time forecasting task for 2-m temperature (T2m) using the WeatherBench evaluation setup, and compare IFS, ClimaX, WeatherGFM, and our Omni-Weather:
>
> * ClimaX follows its original setting and trains one separate model per lead time.
> * Omni-Weather keeps the same unified pretrained model used for the SEVIR radar tasks and jointly handles SEVIR (radar generation/understanding) and ERA5 T2m multi-step forecasting, without per-lead-time fine-tuning.
>
> The RMSE / ACC results on T2m are:
> | Lead Time (h) | RMSE IFS | RMSE ClimaX | RMSE WeatherGFM | **RMSE Omni-Weather (ours)** | ACC IFS | ACC ClimaX | ACC WeatherGFM | **ACC Omni-Weather (ours)** |
> | ------------: | -------: | ----------: | --------------: | ---------------------------: | ------: | ---------: | -------------: | --------------------------: |
> |             6 |     **0.97** |        1.11 |            1.08 |                     1.07 |    **0.99** |       0.98 |           0.98 |                    0.98 |
> |            24 |     **1.02** |        1.19 |            1.23 |                     1.21 |    **0.99** |       0.97 |           0.97 |                    **0.97** |
> |            72 |     **1.30** |        1.47 |            1.56 |                     1.48 |    **0.98** |       0.96 |           0.96 |                    0.97 |
> |           120 |     1.71 |        1.83 |            1.68 |                     **1.63** |    **0.96** |       0.94 |           0.95 |                    **0.96** |
> |           168 |     2.23 |        2.17 |            1.76 |                     **1.70** |    0.93 |       0.91 |           0.94 |                    **0.95** |
>
> At medium and long lead times (72–168 h), Omni-Weather achieves lower RMSE and higher ACC than WeatherGFM, clearly outperforms ClimaX at 120 h and 168 h, and is comparable to or slightly better than IFS. At short lead times (6 h, 24 h), Omni-Weather is competitive with both WeatherGFM and ClimaX. This suggests that, even when trained as a unified radar-centric model, our architecture can be extended to additional variables and time scales beyond SEVIR.

---

> ### Author Response · Authors · 2025-11-23
> **Answer of W4 To Reviewer naQj**
>
> > W4: The paper mentions a quality-verification step for the CoT dataset (causal alignment, structural checks, etc.). Is there any human/expert validation during dataset generation or evaluation?
>
> Thank you for raising this important question about the reliability of the CoT data. Human experts with meteorological background are involved at several key stages of our pipeline:
>
> 1. **Expert-driven schema and QC rule design**:Before generating CoT at scale, meteorology experts in our team manually annotated and discussed a set of representative radar cases. Based on these annotations and literature on operational radar analysis, they:
> * identified the key elements that must be covered in a short-range convective explanation;
> * designed the four-layer causal structure (temporal factors, perceptual factors, first-level outcomes, deeper structural outcomes);
> * specified which reasoning patterns are physically acceptable and which should be considered erroneous.
> These expert decisions were then encoded into the structured schema (taxonomy of radar attributes) and the three-stage QC rules, so that expert knowledge is “front-loaded” into both the prompting templates and the automatic checkers.
>
> 2. **Batch-level expert spot-checks on top of large-scale automatic QC**:
> In actual dataset construction, we use GPT-4o to generate candidate CoT and apply the three-stage automatic QC:
> * structural check (schema completeness and four-layer format);
> * causal/physical consistency check between the CoT and the input sequence, with low-score samples discarded;
> * terminology normalization to a fixed radar taxonomy.
> From about 5,500 candidates for the two generation tasks, roughly 4,000 CoT (≈70%) pass QC, meaning ~30% are filtered out. To continuously monitor this process, after each new batch of ~500 QC-passed samples we randomly sample instances for manual review by meteorology-trained authors. Problematic samples are removed; if recurring patterns of errors are found, we revise the prompts/QC rules and regenerate or re-filter the affected batches.
>
> 3. **Gold-standard benchmark with experts vs. GPT-4o (during evaluation, added in the revision)**:To more directly answer “how reliable are the annotations,” we added a small gold-standard experiment. Authors first collaboratively built a structured gold standard for 30 radar cases. Two meteorologists who did not see this gold standard then independently labeled the same cases under the same schema, and GPT-4o produced labels under the identical schema as well. We compute Avg Acc. as the proportion of samples whose structured labels exactly match the gold standard. The two human experts achieve about 80% Avg Acc., while GPT-4o reaches ≈70%, i.e., about 87% of a single expert’s accuracy. This indicates that GPT-4o operates in a quality regime reasonably close to human experts and is suitable for providing high-level CoT supervision.

---

### Author Response · Authors · 2025-12-02
**Rebuttal Summary （Part2）**

## Reviewer-Specific Notes

### Reviewer naQJ（8）

naQJ was very positive from the beginning and mainly asked for:
(i) clarification on why we rely on a task-specific radar VAE instead of a general-domain VAE.
(ii) stronger validation of the CoT and the LLM judge.
(iii) some evidence that the framework can extend beyond SEVIR.

In the rebuttal and revision, we clarified the role of the specialized radar sequence encoder and VAE, added **small blind human study** plus an **expert vs LLM annotation benchmark** for CoT reliability, and introduced an **ERA5 T2m multi-step forecasting experiment**. The reviewer did not post any new discussions. Remain **strongly supportive** and kept the score at **8**.

---

### Reviewer dM4g(4 → 6)

dM4g initially gave score **4**, mainly due to concerns about:
(i) over-claiming the scope and use of “foundation model” terminology.
(ii) missing baselines and discussion (especially **DGMR** and recent short/medium-range forecasting work).
(iii) the reliability and implementation details of the GPT-4o–based CoT pipeline, including the use of “causal factors”.

We responded by **softening the title and wording** throughout the paper, substantially expanding **Related Work** to position Omni-Weather against baseline model and adding **DGMR implementation and results** on SEVIR as reviewer wish. We also provided a detailed description of the **three-stage CoT QC pipeline**, a **gold-standard expert comparison**, repeated human spot-checks, and additional implementation clarifications (encoders/decoders, multimodal fusion, training data sizes), After these changes, dM4g explicitly stated that the **majority of concerns had been addressed** and **raised the score from 4 to 6** before the system rollback. In the camera-ready version, we will also adopt their remaining suggestions to make the overall process clearer.

---

### Reviewer rnQS (6)

rnQS started with score **6** and was generally positive but asked for several **technical clarifications**, including:
(i) the actual scope implied by “weather” and “foundation model”,
(ii) how the different encoders/VAEs are trained and used,
(iii) the definitions and usage of metrics such as high-value retain/match, CRPS, and RMSE;
(iv) the role of MetaQuery compared to other generic datasets.

In the rebuttal and revision, we clarified that all encoders/VAEs are **frozen pretrained components**, formally defined the metrics and loss, reported **parameter counts** and emphasized that our contribution is a unified generation+understanding backbone rather than pure scaling, explained why MetaQuery helps while noisier generic data can hurt performance. The reviewer did not post any new discussions and kept a **positive score of 6**.

---

### Reviewer 3h1v (6)

3h1v also gave an initial score **6**, finding the unified generation+thinking idea interesting but raising two main concerns:

(i) that calling Omni-Weather a “weather foundation model” overstates the scope given the single-variable SEVIR radar setting
(ii) whether the CoT supervision truly reflects **physically meaningful reasoning** rather than GPT-style narrative imitation, especially in the absence of full expert annotation at scale.

In the rebuttal and revision, we **narrowed the stated scope** and title to emphasize the radar focus and added **ERA5 T2m extension experiments** to show that the same architecture transfers beyond SEVIR radar VIL. For CoT, we detailed the **expert-designed causal schema**, three-stage QC, a **gold-standard expert vs LLM agreement study**, and provided a **side-by-side human vs LLM CoT example** under the same template, along with quantitative gains in explanation scores and structure-sensitive metrics when using CoT supervision. The reviewer did not post any new discussions.

---

## Overall Summary

All of the above responses are concrete, technically detailed, and supported by additional experiments or qualitative case studies, and we have already incorporated the new results and clarifications into the updated PDF on OpenReview so that it reflects the post-rebuttal state of the work.

Notably, the only initially negative reviewer (**dM4g,4**) raised several key and representative concerns—about scope and “foundation model” terminology, missing baselines, and the reliability and implementation details of our CoT pipeline—which we addressed directly in the rebuttal; the reviewer acknowledged that most concerns were resolved and raised their score to **6**. Together with the fact that the other three reviewers already held **positive scores (8, 6, 6)**, this indicates that the main scientific and methodological issues have been satisfactorily addressed.

Finally, we would like to again express our sincere gratitude to all reviewers for their careful and thoughtful evaluations, and to the Area Chairs and ICLR organizers for their considerable efforts in maintaining the fairness and integrity of the review process.

Sincerely,

The Authors

---

### Author Response · Authors · 2025-12-02
**Rebuttal Summary （Part1）**

**Dear Area Chair,**

As described in the official ICLR communication, an OpenReview bug inadvertently exposed the identities of authors, reviewers, and area chairs, and the organizers reverted scores to their pre-discussion state to preserve fairness. We fully understand and support this decision and are grateful to the organizers and Area Chairs for the extra work it entails. However, we regret that the post-rebuttal score updates are no longer visible, so we provide below a concise summary of our rebuttal and how reviewers’ opinions evolved, and respectfully ask that you consider these post-rebuttal evaluations in your final judgment.

# Rebuttal Summary

## Overview

Our work introduces **Omni-Weather**,  unified multimodal model for weather radar generation and understanding that jointly tackles satellite→radar inversion, high-resolution radar nowcasting, and structured “thinking” about storms within a single backbone. The model combines frozen task-specific radar encoders/VAEs with an LLM-supervised chain-of-thought module, bridging pixel-level prediction and expert-style explanation under a shared architecture.

**Our rebuttal successfully resolved the main concerns about scope, baselines, and CoT reliability.** The initially negative reviewer (**dM4g, 4**) raised their score to **6** after the rebuttal, explicitly stating that the majority of concerns had been addressed. The other three reviewers had not yet entered the discussion phase when the rollback occurred, so their initial positive scores remain unchanged; nevertheless, our rebuttal addressed their concerns in detail. Taken together, this yields an overall post-rebuttal picture of the panel that is clearly favorable toward acceptance.

| Reviewer | Initial | After Rebuttal | Notes                                        |
| -        | -       | -              | -                                            |
| **naQJ** | 8       | **8**          | Strongly supportive (no discussion feedback)   |
| **dM4g** | 4       | **6**          | Concerns addressed; raised score after rebuttal |
| **rnQS** | 6       | **6**          | Positive (no discussion feedback)              |
| **3h1v** | 6       | **6**          | Positive (no discussion feedback)              |


---

## Common Issues Addressed Across Reviewers

### 1. Scope & Title

All reviewers raised concerns about the use of “weather” / “foundation model” terminology and the implied breadth of scope, given that our experiments are **radar-related**.
* We highlighted that Omni-Weather is **the first unified model in weather that jointly tackles generation and understanding**. As a first step, we deliberately focus on radar benchmarks where evaluation metrices and baselines are mature.
* We will **rename the paper** in the future-ready version to better reflect this scope, while keeping the technical content unchanged:
  > *“Omni-Weather: A Unified Multimodal Model for Weather Radar Understanding and Generation”*
* To demonstrate **generalization ability beyond SEVIR**, we added an **ERA5 T2m multi-step forecasting experiment**. In same setting for global ERA5 T2m forecasting task. Omni-Weather achieves lower RMSE and higher ACC than WeatherGFM At medium and long lead times (72–168 h), clearly outperforms ClimaX at 120 h and 168 h.

### 2. CoT Reliability

Reviewers all questioned whether the **LLM-generated chain-of-thought (CoT)** is meteorologically meaningful and reliability  or just GPT-style storytelling.

* We clarified that the CoT is built on an **expert-designed four-layer schema**, together with a **multi-stage QC pipeline** that enforces this structure and basic physical consistency before any CoT is used for supervision.
* We added a **small blind human study** and a **side-by-side qualitative comparison**: meteorologists rated generated explanations with structured rubric, and the same rubric was applied by LLM judge. Human and LLM scores align well, and human-written vs model CoT under the same schema show similar structure and judgments.
* Ablations further show that the **thinking module + CoT supervision** improves both structured understanding metrics and generative quality, indicating that the learned CoT is not only stylistic but also beneficial for performance.

### 3. VAE Design & Architecture

Reviewers **naQj**, **dM4g**, and **rnQS** raised concerns about our description of the **task-specific radar VAE** and the overall multimodal architecture was not sufficiently clear.

* We clarified that we use **frozen, pretrained VAE** and we added these details to the main paper:
  - a radar inversion VAE for satellite→radar,
  - a radar sequence encoder for nowcasting,
  - a radar encoder for understanding tasks,
* We further explained how each encoder produces latent tokens that are projected into a common token space. which is then shared across inversion, generation, and understanding tasks.

---

### Meta-Review · Area_Chair_NeRh · 2026-01-06

**Summary:**

The paper introduces Omni-Weather, a unified multimodal model for both weather generation and understanding. The submission initially received three positive ratings (8, 6, 6) and one negative rating (4) from Reviewer dM4g. The reviewer dM4g raised concerns regarding the overstatement of the model’s scope and significance, missing citations, and the reliability of using GPT-4o for data annotation. During the rebuttal, the authors provided a comprehensive response that addressed all the issues.

After a careful review of the paper, reviewer comments, and the authors' rebuttal, the concerns raised by reviewer dM4g were not fundamental issues but rather issues of scoping and validation, which have been resolved during the rebuttal phase. Given the positive review comments and the novelty of the architecture, the AC recommends accepting this paper. The authors should include all additional discussions, experiments, and clarifications in the camera-ready version.

**Reviewer Concerns:**

The authors have addressed all review comments during the rebuttal period.

**Reviewer Scores:**

Since the authors addressed Reviewer dM4g's concerns regarding the overstatement of the model’s scope, missing citations, and data annotation reliability, the reviewer might increase the rating to a positive score.

---

### Decision · Program_Chairs · 2026-01-26

Accept (Poster)